# Fat2 polarizes the WAVE complex in *trans* to align cell protrusions for collective migration

**Audrey Miller Williams[1], Seth Donoughe[1], Edwin Munro[1,2,3], Sally Horne-Badovinac[1,2]\***

[1]Department of Molecular Genetics and Cell Biology, University of Chicago, Chicago, United States; [2]Committee on Development, Regeneration, and Stem Cell Biology, University of Chicago, Chicago, United States; [3]Institute for Biophysical Dynamics, University of Chicago, Chicago, United States

**Abstract** For a group of cells to migrate together, each cell must couple the polarity of its migratory machinery with that of the other cells in the cohort. Although collective cell migrations are common in animal development, little is known about how protrusions are coherently polarized among groups of migrating epithelial cells. We address this problem in the collective migration of the follicular epithelial cells in *Drosophila melanogaster*. In this epithelium, the cadherin Fat2 localizes to the trailing edge of each cell and promotes the formation of F-actin-rich protrusions at the leading edge of the cell behind. We show that Fat2 performs this function by acting in trans to concentrate the activity of the WASP family verprolin homolog regulatory complex (WAVE complex) at one long-lived region along each cell's leading edge. Without Fat2, the WAVE complex distribution expands around the cell perimeter and fluctuates over time, and protrusive activity is reduced and unpolarized. We further show that Fat2's influence is very local, with sub-micron-scale puncta of Fat2 enriching the WAVE complex in corresponding puncta just across the leading-trailing cell-cell interface. These findings demonstrate that a trans interaction between Fat2 and the WAVE complex creates stable regions of protrusive activity in each cell and aligns the cells' protrusions across the epithelium for directionally persistent collective migration.

**\*For correspondence:**
shorne@uchicago.edu

**Competing interest:** The authors declare that no competing interests exist.

## Editor's evaluation

This paper addresses a fundamental aspect of cell migration, how the direction of cell migration is established. It links molecules involved in planar polarity to the migration machinery using quantitative imaging techniques capitalizing on the *Drosophila* genetic tool box. It adds to our growing knowledge of how collective cell migration is regulated and introduces an exciting new line of inquiry.

## Introduction

Collective cell migration is essential for a variety of morphogenetic processes in animals (*Friedl and Gilmour, 2009*; *Scarpa and Mayor, 2016*; *Norden and Lecaudey, 2019*; *Perez-Vale and Peifer, 2020*). As with individual cell migrations, adherent collective migrations are driven by the concerted action of cell protrusions, contractile actomyosin networks, and adhesions to a substrate (*Scarpa and Mayor, 2016*; *Bodor et al., 2020*; *Buttenschön and Edelstein-Keshet, 2020*). To move forward, individual cells polarize these structures along a migratory axis, and to move persistently in one direction, they need to maintain that polarity stably over time (*Stock and Pauli, 2021*). Collective cell migrations

**Figure 1.** Introduction to egg chamber rotation and follicle cell protrusions. (**A**) Diagram of a stage 6 egg chamber in cross-section. Anterior is left, posterior right. (**B**) Three-dimensional diagram of an egg chamber with the anterior half shown. Arrows indicate the migration of follicle cells along the basement membrane and the resulting rotation of the egg chamber around its anterior-posterior axis. (**C**) Diagram of three follicle cells. Their apical surfaces adhere to the germ cells and their basal surfaces adhere to the basement membrane. The dashed line represents the basal imaging plane used throughout this study except where indicated. (**D**) Images of the leading edges of two cells expressing Ena-GFP and WAVE complex label Abi-mCherry, and with F-actin stained with phalloidin. (**E**) Diagrams showing the organization of F-actin and its regulators at the leading edge. The WAVE complex builds a lamellipodial actin network, within which Ena builds filopodia. (**F**) Images of F-actin (phalloidin) and cell interfaces (anti-Discs Large) in control, *ena*-RNAi, and *abi*-RNAi backgrounds. Expression of *ena*-RNAi strongly depletes filopodia, revealing the less-prominent lamellipodial actin network, whereas *abi*-RNAi expression removes both filopodia and lamellipodia.

introduce a new challenge: to move together, the group of migrating cells must be polarized in the same direction (*Stock and Pauli, 2021*). Otherwise, they would exert forces in different directions and move less efficiently, separate, or fail to migrate altogether.

The epithelial follicle cells of the *Drosophila melanogaster* ovary are a powerful experimental system in which to investigate how local interactions among migrating cells establish and maintain group polarity. Follicle cells are arranged in a continuous, topologically closed monolayer epithelium that forms the outer cell layer of the ellipsoidal egg chamber—the organ-like structure that gives rise to the egg (*Duhart et al., 2017*; *Figure 1A–C*). The apical surfaces of follicle cells adhere to a central germ cell cluster, and their basal surfaces face outward and adhere to a surrounding basement membrane extracellular matrix. The follicle cells migrate along this stationary basement membrane, resulting in rotation of the entire cell cluster (*Haigo and Bilder, 2011*). As the cells migrate, they secrete additional basement membrane proteins (*Haigo and Bilder, 2011*). The coordination of migration with secretion causes the cells to produce a basement membrane structure that channels tissue growth along one axis (*Gutzeit et al., 1991*; *Haigo and Bilder, 2011*; *Isabella and Horne-Badovinac, 2016*; *Crest et al., 2017*). Follicle cell migration lasts for roughly 2 days, and the migration

direction—and resulting direction of egg chamber rotation—is stable throughout (*Chen et al., 2017*; *Stedden et al., 2019*). The edgeless geometry of the epithelium means cells are not partitioned into 'leader' and 'follower' roles, and there is no open space, chemical gradient, or other external guidance cue to dictate the migration direction. Instead, this feat of stable cell polarization and directed migration is accomplished through local interactions between the migrating cells themselves (*Barlan et al., 2017*; *Stedden et al., 2019*).

Follicle cell migration is driven, in part, by lamellipodial protrusions that extend from the leading edge of each cell (*Gutzeit et al., 1991*; *Cetera et al., 2014*). Lamellipodia are built by the WASP family verprolin homolog regulatory complex (WAVE complex) (*Miki et al., 1998*; *Miki et al., 2000*), which is a protein assembly composed of five subunits: SCAR/WAVE, Abi, Sra1/Cyfip, Hem/Nap1, and HSPC300 (*Chen et al., 2010*). The WAVE complex adds branches to actin filaments by activating the Actin-related proteins-2/3 complex (Arp2/3) and elongates existing filaments, building the branched actin network that pushes the leading edge forward (*Machesky et al., 1999*; *Bieling et al., 2018*; *Mullins et al., 2018*). Embedded within the lamellipodia are Enabled (Ena)-dependent filopodia, which are visually prominent with F-actin labeling but dispensable for migration (*Cetera et al., 2014*; *Figure 1D and E*). Removal of filopodia reveals the underlying lamellipodial actin network, whereas removal of WAVE complex subunits eliminates all protrusive structures (*Cetera et al., 2014*; *Figure 1F*). We use the term 'protrusions' to encompass both of these F-actin networks and the membrane deformations they cause.

The follicle cells align their protrusions across the tissue, a form of planar polarity (*Gutzeit et al., 1991*; *Cetera et al., 2014*). The atypical cadherin Fat2 is required both for this planar polarity and for collective migration to occur (*Viktorinová et al., 2009*; *Viktorinová and Dahmann, 2013*; *Horne-Badovinac, 2017*). Fat2 is planar-polarized to the trailing edge of each cell (*Viktorinová and Dahmann, 2013*), where it promotes the formation of protrusions at the leading edge of the cell immediately behind (*Barlan et al., 2017*). Interestingly, in addition to migration depending on polarized Fat2 activity, Fat2's planar polarity also depends on epithelial migration (*Barlan et al., 2017*). It is not known how Fat2 regulates lamellipodia or cell polarity, or how these processes influence one another. We hypothesized that Fat2 acts as a coupler between tissue planar polarity and cell protrusion by polarizing WAVE complex activity to the leading edge of each cell. To test this, we used genetic mosaic analysis and quantitative imaging of fixed and live tissues to dissect Fat2's contributions to protrusivity and protrusion polarity at cell and tissue scales.

We show that Fat2 signals in trans, entraining WAVE complex activity to one long-lived region along each cell's leading edge. Without Fat2, the WAVE complex accumulates transiently at different regions around the cell perimeter, and cell protrusivity is reduced and unpolarized. The interaction between Fat2 and the WAVE complex is non-cell-autonomous but very local, with sub-micron-scale puncta of Fat2 along the trailing edge concentrating the WAVE complex just across the cell-cell interface, at the tips of filopodia embedded within the lamellipodium. These findings demonstrate how an intercellular interaction between Fat2 and the WAVE complex promotes cell protrusivity, stabilizes regions of protrusive activity along the cell perimeter, and aligns protrusions across the epithelium by coupling leading and trailing edges. Fat2-WAVE complex interaction thereby stabilizes the planar polarity of protrusions for directionally persistent collective migration.

## Results

### Fat2 increases and polarizes protrusions at the basal surface of the follicular epithelium

Recent work has shown that Fat2 regulates migration of the follicular epithelium by polarizing F-actin-rich protrusions; specifically, Fat2 at the trailing edge of each cell causes protrusions to form at the leading edge of the cell behind it, and without Fat2, protrusions are reduced or lost (*Squarr et al., 2016*; *Barlan et al., 2017*). Beyond this qualitative description, it is not known how Fat2 modulates cell protrusion.

We sought to obtain a deeper, time-resolved view of the role of Fat2 in regulating protrusivity and protrusion distribution. To do so, we developed methods to segment cell membrane extensions and measure their lengths and orientations, and applied these methods to timelapse movies of the basal surface of control and *fat2^N103-2^* epithelia (a null allele, hereafter referred to as *fat2*; *Figure 2*,

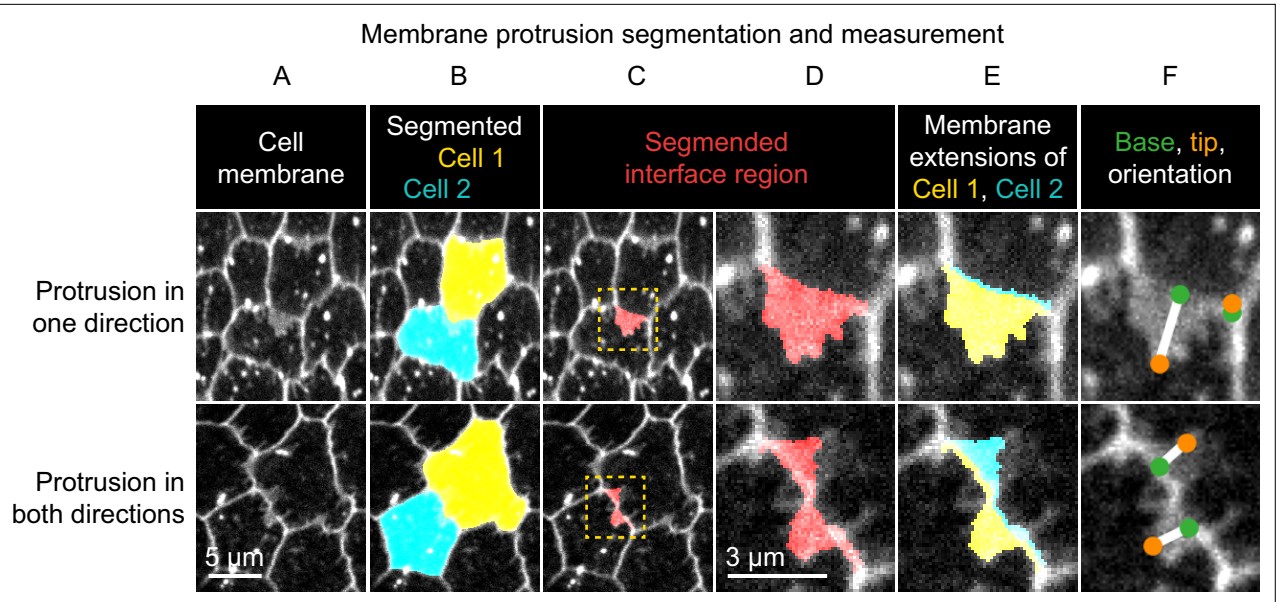

**Figure 2.** Method used to segment and measure membrane protrusions. Top row shows an example of a pair of neighboring cells in which one cell is protruding across their shared interface. Bottom row shows a case in which both cells are protruding across the interface. (**A**) Cell interfaces and protrusions were labeled with a membrane dye and timelapses of the basal surface were collected. (**B**) Cells were automatically segmented with a watershed-based method, and segmentation errors were hand-corrected. (**C**) The bright interface region between each pair of neighboring cells was identified using a watershed-based method. This region includes the interface and any membrane protrusions that extend across it. (**D**) An enlargement of the boxed regions of (**C**). (**E**) The interface region was divided into two parts by the shortest path from vertex to vertex within the region, which approximates the true cell-cell interface position. The two resulting regions were then assigned to the cell from which they each extended. The area of these regions and the length of the interface between them were used to define average membrane extension length (as described in Materials and methods). (**F**) The tip and base of each region were identified, and then used to measure lengths and orientations (see Materials and methods).

*Figure 3*). A detailed description of the segmentation approach is included in the Materials and Methods. To analyze these data, we first measured the average lengths of membrane extensions from all cell-cell interfaces (*Figure 3A and B*). The distribution of measured lengths was unimodal, with no natural division between protrusive and non-protrusive interfaces. Therefore, to establish an empirically grounded cutoff between these categories, we recorded timelapse movies of control epithelia treated with the Arp2/3 inhibitor CK-666, which are non-migratory and almost entirely non-protrusive (*Cetera et al., 2014*). We used measurements from CK-666-treated epithelia to set a cutoff for the minimum length of a protrusion: any edges with membrane extensions longer than the 98th percentile of those in CK-666-treated epithelia were considered *protrusive* for subsequent analysis (*Figure 3B*).

Using this quantification approach, we first asked how tissue protrusivity was affected by the loss of Fat2. We found that the protrusivity of *fat2* epithelia was lower than that of control epithelia on average, but highly variable, with overlap between the protrusivity distributions of both untreated and CK-666-treated epithelia (*Figure 3B and C*; *Figure 3—figure supplement 1A,B*; *Figure 3—video 1*). As a complementary method, we also measured protrusivity via F-actin labeling in fixed and live tissues, using *abi*-RNAi-expressing epithelia as a nearly non-protrusive benchmark. The results largely paralleled those seen with membrane labeling (*Figure 3—figure supplement 2*; *Figure 3—video 2*); however, the disparity in protrusivity between *fat2* and control epithelia appeared larger when measured using F-actin labeling than when measured with membrane labeling (*Figure 3C*; *Figure 3—figure supplement 2*). Images of follicle cell protrusions visualized by F-actin staining are dominated by fluorescence from filopodia (*Figure 1F*). The appearance of lower protrusivity of *fat2* epithelia as measured with an F-actin label may therefore indicate that filopodia are disproportionately reduced by loss of Fat2. Altogether, these data show that *fat2* epithelia are less protrusive than control epithelia, but do retain some protrusive activity.

These results raised an important question—if some *fat2* epithelia have levels of membrane protrusivity comparable to that of control epithelia, then why do all *fat2* epithelia fail to migrate (*Viktorinová and Dahmann, 2013*; *Chen et al., 2017*; *Barlan et al., 2017*)? We hypothesized that

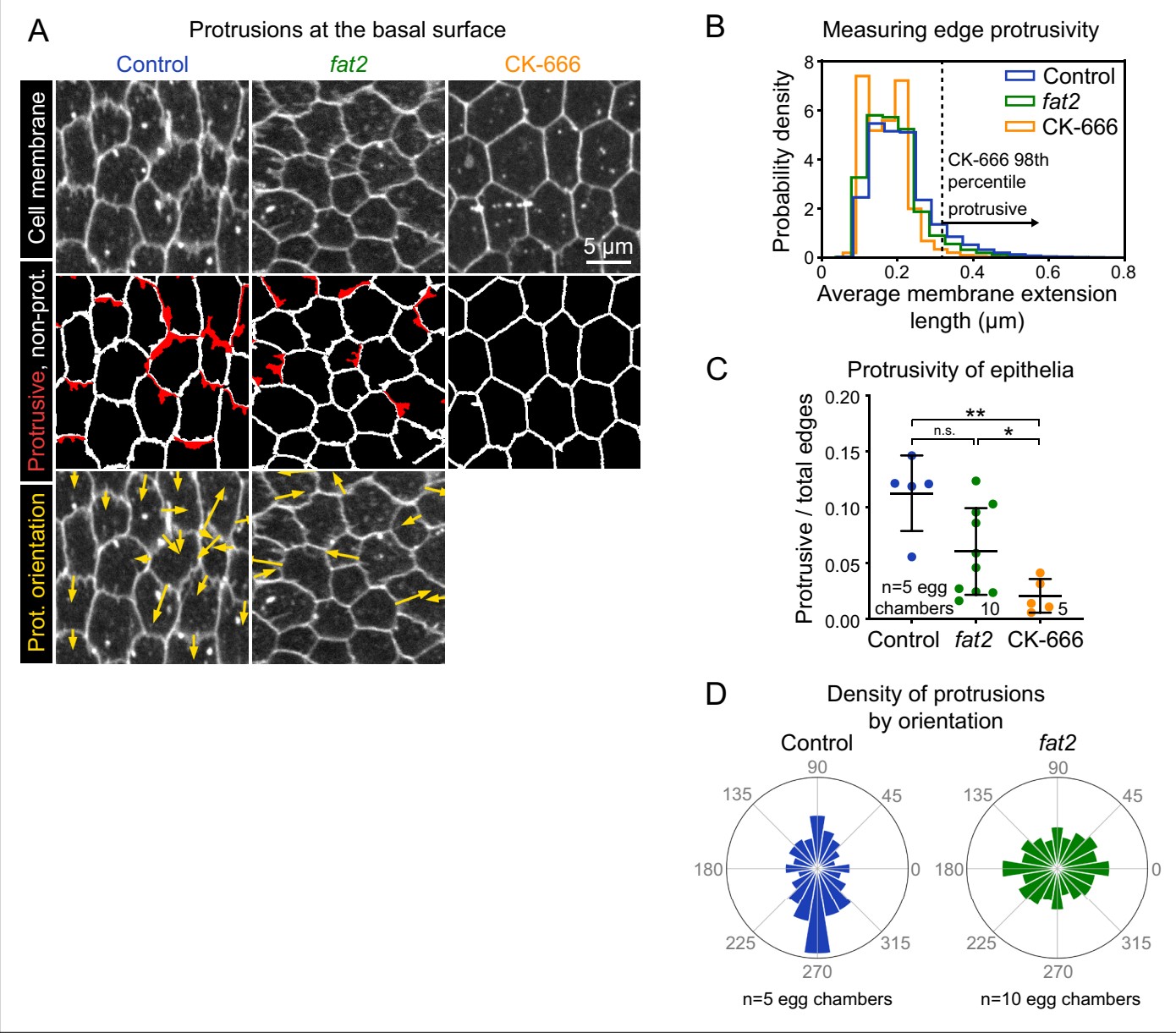

**Figure 3.** Fat2 increases and polarizes follicle cell protrusivity. (**A**) Timelapse frames of control, *fat2*, and CK-666-treated epithelia labeled with a membrane dye. Middle row shows segmented edges. Protrusive edges, defined as edges with average membrane extension lengths longer than the 98th percentile of those of CK-666-treated epithelia, are shown in red. Non-protrusive edges are white. Bottom row shows arrows indicating the orientation of each protrusion overlaid on labeled cell membrane. Arrows originate at protrusion bases and have lengths proportional to protrusion lengths. See related *Figure 3—video 1* and *Figure 3—video 3*. (**B**) Histogram showing the distribution of average membrane extension lengths. The 98th percentile length threshold for CK-666-treated epithelia is indicated. (**C**) Plot showing the ratio of protrusive to total edges. The protrusivity of *fat2* epithelia is variable, with a distribution overlapping with control and CK-666-treated epithelia. Welch's ANOVA (W(2,9.3)=15.89, p=0.0012) with Dunnet's T3 multiple comparisons test; n.s. p=0.07, *p=0.04, **p=0.004. Bars indicate mean ± SD. Counts of protrusive and total edges are listed in *Figure 3— source data 1*. See *Figure 3—figure supplement 1* for alternate measurements of protrusivity. (**D**) Polar histograms of the distribution of protrusion orientations in control and *fat2* epithelia. Anterior is left, posterior is right, and in control epithelia images were flipped as needed so that migration is always oriented downward. Bar areas scale with the fraction of protrusions. Protrusion counts are listed are in *Figure 3—source data 1*. Control protrusions point predominantly in the direction of migration, whereas *fat2* protrusions are less polarized. Histograms from individual epithelia can be found in *Figure 3—figure supplement 1*.

The online version of this article includes the following video, source data, and figure supplement(s) for figure 3:

**Figure 3—video 1.** Membrane protrusivity of control, *fat2*, and CK-666-treated epithelia.
https://elifesciences.org/articles/78343/figures#fig3video1

*Figure 3 continued on next page*

*Figure 3 continued*

**Source data 1.** Sample sizes of dataset used to generate plots in *Figure 3B-D*, *Figure 3—figure supplement 1*, and Figure 7C.

**Figure 3—video 2.** F-actin protrusivity and protrusion polarity of control and *fat2* epithelia.

https://elifesciences.org/articles/78343/figures#fig3video2

**Figure 3—video 3.** Protrusion orientation in control and *fat2* epithelia.

https://elifesciences.org/articles/78343/figures#fig3video3

**Source data 2.** Membrane protrusivity of control, fat2, and CK-666-treated epithelia.

**Figure supplement 1.** Membrane extension length and protrusion orientation in individual egg chambers.

**Figure supplement 2.** Actin protrusions are reduced and unpolarized without Fat2 and further reduced without the WAVE complex.

**Figure supplement 2—source data 1.** F-actin enrichment at cell-cell interfaces of control, fat2, and abi-RNAi-expressing epithelia.

**Figure supplement 2—source data 2.** F-actin interface enrichment by angle in control and fat2 epithelia.

**Figure supplement 2—source data 3.** F-actin enrichment at leading-trailing interfaces in control and fat2 epithelia.

the mispolarization of protrusions across the tissue contributes to *fat2* migration failure. In control epithelia, the majority of protrusions were polarized in the direction of migration, orthogonally to the egg chamber's anterior-posterior axis (*Figure 3A and D*; *Figure 3—figure supplement 1C*; *Figure 3— video 3*). In contrast, in *fat2* epithelia, protrusions were fairly uniformly distributed in all directions or biased in two opposite directions (*Figure 3A and D*; *Figure 3—figure supplement 1C*; *Figure 3— video 3*). Where an axial bias was present, the axis was inconsistent between egg chambers. We also confirmed this finding using F-actin labeling of protrusions. To compare the planar polarity of F-actin protrusions between control and *fat2* epithelia, we measured F-actin enrichment at cell-cell interfaces as a function of the angle of the interface relative to the egg chamber's anterior-posterior axis. We again saw that protrusions were planar-polarized in control epithelia and unpolarized in *fat2* epithelia (*Figure 3—figure supplement 2*). These data show that Fat2 is required to polarize protrusions in a common direction across the epithelium.

Because Fat2 regulates both follicle cell migration and planar polarity, and migration and planar polarity are interdependent (*Viktorinová et al., 2009*; *Viktorinová and Dahmann, 2013*; *Cetera*

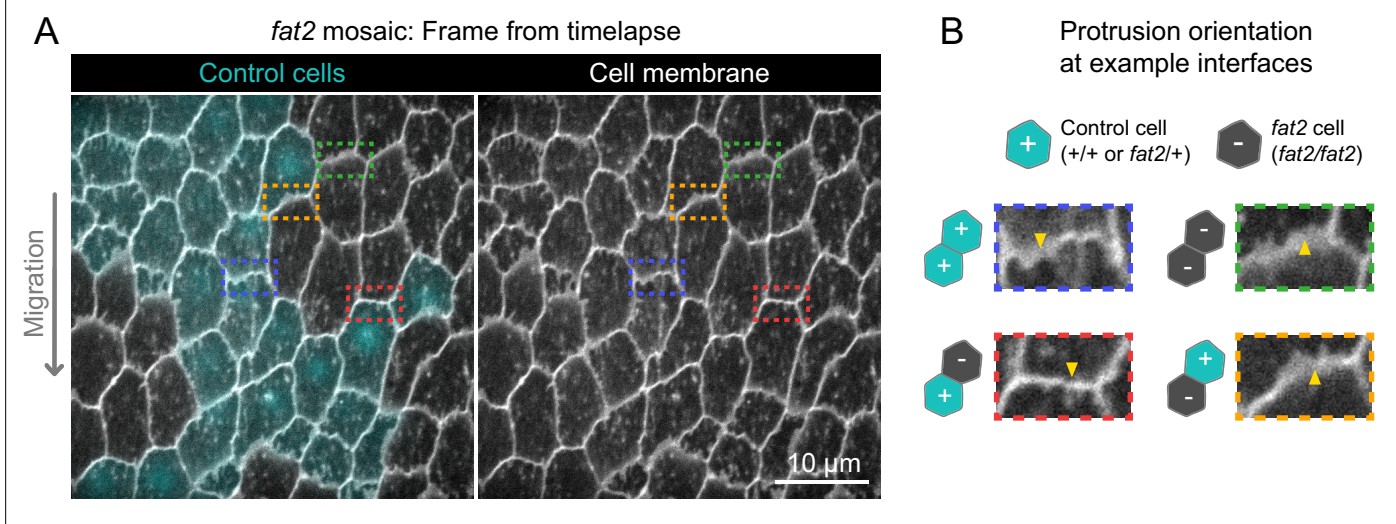

**Figure 4.** Fat2 acts locally across the cell interface to orient membrane protrusions. (**A**) Timelapse frame of a *fat2* mosaic epithelium with cell membrane labeled, used to evaluate protrusion orientations in control or *fat2* cells within a migratory context. Boxes indicate examples of leading-trailing interfaces between neighbor pairs with each possible combination of genotypes. Representative of 9 similar timelapse movies. See related *Figure 4—video 1*. (**B**) Larger images of the interfaces boxed in (**A**) showing that protrusions are misoriented when *fat2* cells are ahead of the interface regardless of the genotype of the cell behind the interface. Arrows point in the direction of protrusion.

The online version of this article includes the following video for figure 4:

**Figure 4—video 1.** Membrane protrusion in a fat2 mosaic epithelium.

https://elifesciences.org/articles/78343/figures#fig4video1

*et al., 2014*; *Barlan et al., 2017*), the unpolarized protrusions of *fat2* epithelia could be a cause or a consequence of inability of *fat2* epithelia to migrate. To distinguish between these possibilities, we exploited the fact that small groups of *fat2* cells can be carried along by neighboring non-mutant, migratory cells (*Viktorinová and Dahmann, 2013*), allowing us to evaluate polarity of protrusions from *fat2* cells in a migratory context. We generated *fat2* mosaic tissues that had sufficiently small fractions of mutant cells that the tissue as a whole still migrated, and found that *fat2* cells in these tissues were often protrusive, but their protrusions were not polarized in the direction of migration (*Figure 4*; *Figure 4—video 1*). This demonstrates that Fat2 does not simply polarize protrusions indirectly by maintaining tissue-wide migration. Rather, Fat2 is required at the scale of groups of cells to polarize those cells' protrusions in alignment with the direction of collective migration.

## Fat2 increases and polarizes the WAVE complex at the basal surface of the follicular epithelium

Follicle cell protrusions are built by the WAVE complex (*Cetera et al., 2014*), which commonly acts in a circuit with active Rac and PI(3,4,5)P3. We hypothesized that Fat2 polarizes protrusions by polarizing the distribution of one of these circuit components. To visualize their activity, we focused on the WAVE complex, whose localization most closely determines and reports sites of protrusion. Using CRISPR/Cas9, we endogenously tagged the WAVE complex subunit Sra1 with eGFP (hereafter: Sra1-GFP), allowing us to visualize its localization and dynamics at endogenous levels.

We confirmed that Sra1-GFP flies are viable and fertile when the tagged allele is homozygous, Sra1-GFP localizes to follicle cell leading edges like other WAVE complex labels (*Cetera et al., 2014*; *Squarr et al., 2016*), its localization depends on WAVE complex subunit Abi, and F-actin protrusions appear normal (*Figure 5A and B*; *Figure 5—figure supplement 1A-C*). Migration was slower when Sra1-GFP was present in two copies (*Figure 5—figure supplement 1D*), so we performed all subsequent experiments with one copy of Sra1-GFP.

With an endogenous WAVE complex label in hand, we investigated how Fat2 affects WAVE complex localization. Previous work has shown that WAVE complex levels are reduced at the basal surface of follicle cells lacking Fat2 (*Squarr et al., 2016*). Consistent with this result, we found that Sra1-GFP levels were lower along cell-cell interfaces at the basal surface of *fat2* epithelia than of control epithelia (*Figure 5—figure supplement 2*). Planar polarity of Sra1-GFP across the epithelium was also lost in the absence of Fat2 (*Figure 5—figure supplement 2*). Fat2 acts non-cell-autonomously to cause protrusions to form at the leading edge of the cell just behind (*Barlan et al., 2017*; *Figure 5C*), so we next tested the hypothesis that Fat2 localizes the WAVE complex to the leading edge in the same non-cell-autonomous pattern. We did this using *fat2* mosaic epithelia, in which we could measure Sra1-GFP levels at leading-trailing interfaces shared by control and *fat2* cells. We found that Sra1-GFP levels were normally enriched along the leading edges of *fat2* cells if control cells were present immediately ahead, showing that Sra1 can still localize to the leading edge of cells lacking Fat2. Conversely, Sra1-GFP levels were reduced along the leading edge of control cells if *fat2* cells were immediately ahead (*Figure 5D and E*; *Figure 5—figure supplement 2*). We also observed a corresponding non-autonomous pattern of membrane protrusion polarity in timelapse movies of *fat2* mosaic epithelia (*Figure 4*; *Figure 4—video 1*). We conclude that Fat2 acts non-cell-autonomously to localize the WAVE complex to leading edges, resulting in tissue-wide planar polarization of protrusive activity, and thereby in collective cell migration.

We next asked if, by recruiting the WAVE complex to the leading edge, Fat2 was reducing its levels at other membrane sites, and thereby suppressing mispolarized protrusion. To test whether Fat2 was measurably depleting the non-leading-edge WAVE complex pool, we compared the level of Sra1-GFP at the medial basal surfaces of cells in control or *fat2* epithelia, or between control and *fat2* cells in mosaic epithelia (see diagram in *Figure 5F*). In both cases, we measured small increases in mean Sra1-GFP in *fat2* cells compared to control cells, but these were not statistically significant (*Figure 5D and F*; *Figure 5—figure supplement 2*). Our measurements may not be sensitive enough to detect redistribution of Sra1-GFP occurring across a broad membrane area, or the Sra1-GFP population may be redistributing away from the basal surface. It therefore remains to be determined whether by concentrating the WAVE complex at the leading edge, Fat2 also depletes the WAVE complex from other membrane sites, and thereby suppresses mispolarized protrusions.

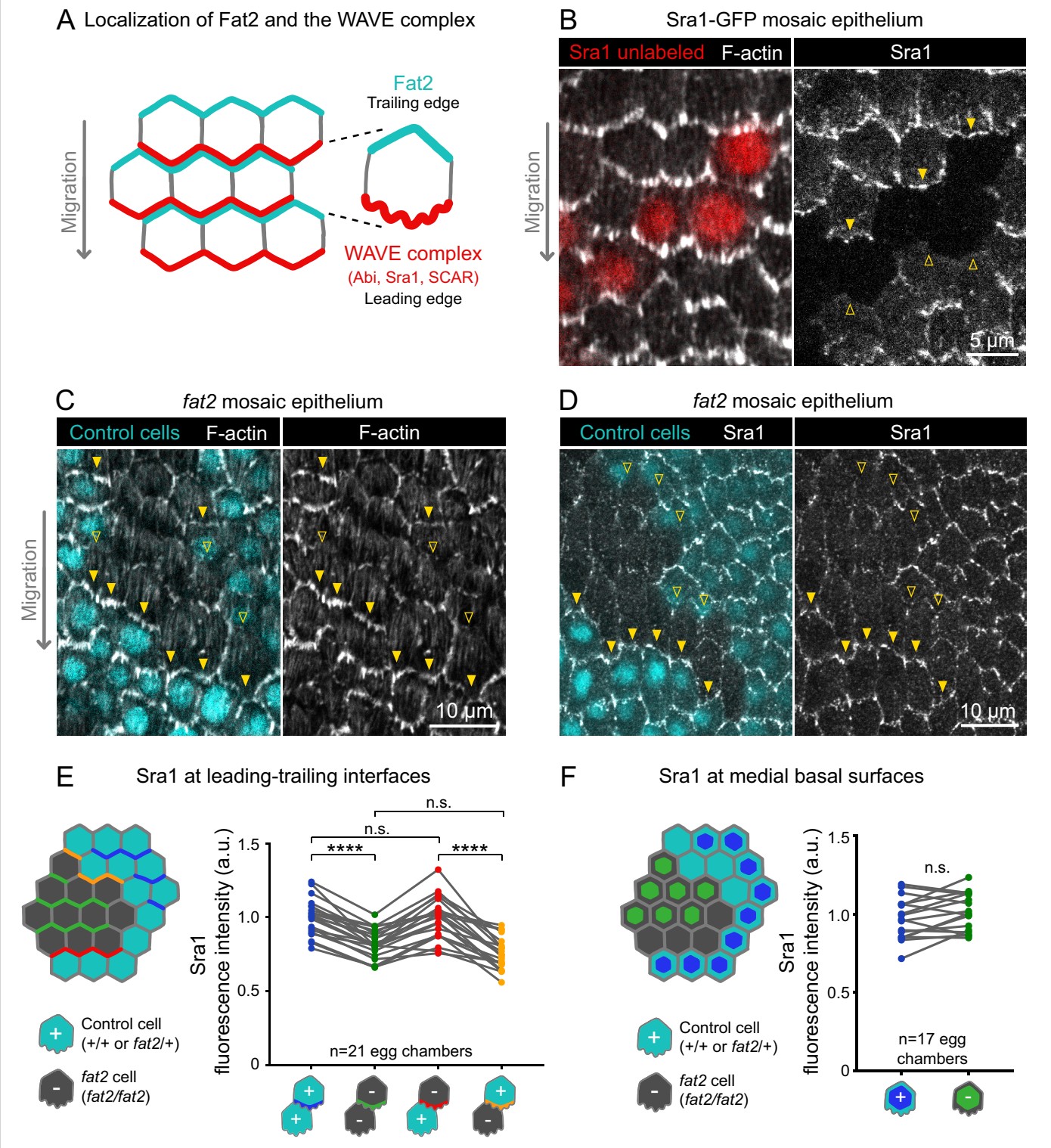

**Figure 5.** Fat2 in each cell concentrates the WAVE complex at the leading edge of the cell behind. (**A**) Diagram showing Fat2 localization at the trailing edge and WAVE complex at the leading edge of the basal surface of follicle cells. The WAVE complex subunits referenced in this paper listed. (**B**) Images of an Sra1-GFP mosaic epithelium with phalloidin-stained F-actin, showing Sra1-GFP enrichment at leading edges (filled arrows) and not trailing edges (open arrows). (**C**) Images of a *fat2* mosaic epithelium with phalloidin-stained F-actin. Filled arrows indicate leading edges of *fat2* cells behind control cells, where protrusions are present. Open arrows indicate leading edges of control cells behind *fat2* cells, where protrusions are reduced. (**D**) Images of a *fat2* mosaic epithelium expressing Sra1-GFP. Filled arrows indicate leading edges of *fat2* cells behind control cells. Open arrows indicate

*Figure 5 continued on next page*

*Figure 5 continued*

leading edges of control cells behind *fat2* cells. (**E,F**) Quantification of Sra1-GFP mean fluorescence intensity in *fat2* mosaic epithelia along leading-trailing interfaces (**E**) or medial basal surfaces (**F**) Diagrams to the left of plots show the measured regions with respect to control (cyan) and *fat2* (gray) cells. The genotype(s) of cells in each measured category are shown below the x-axis. Lines connect measurements from the same egg chamber. (**E**) Sra1-GFP is reduced at the leading edge of cells of any genotype behind *fat2* cells. Repeated measures ANOVA [F(3,80)=22.77, p<0.0001] with post-hoc Tukey's test; n.s. (left to right) p=0.99, 0.24, ****p<0.0001. (**F**) Sra1-GFP is not significantly changed at the medial basal surface of *fat2* cells. Paired t-test; n.s. p=0.08.

The online version of this article includes the following source data and figure supplement(s) for figure 5:

**Source data 1.** Sra1 levels at the basal surface of *fat2* mosaic epithelia.

**Figure supplement 1.** Evaluation of endogenous Sra1-GFP functionality.

**Figure supplement 1—source data 1.** Migration speeds of epithelia expressing Sra1-GFP.

**Figure supplement 2.** Fat2 concentrates the WAVE complex at cell-cell interfaces and polarizes it across the epithelium.

**Figure supplement 2—source data 1.** Sra 1 enrichment at cell-cell interfaces of control and fat2 epithelia.

**Figure supplement 2—source data 2.** Sra 1 interface enrichment by angle in control and fat2 epithelia.

**Figure supplement 2—source data 3.** Sra 1 enrichment at leading-trailing interfaces in control and fat2 epithelia.

**Figure supplement 2—source data 4.** Sra 1 levels at the basal surface of cells in fat2 mosaic epithelia.

## Fat2 stabilizes a region of WAVE complex enrichment and protrusivity in *trans*

In individually migrating cells, the excitable dynamics of the WAVE complex and its regulators enable it to form transient zones of enrichment along the cell perimeter even in the absence of a directional signal (*Weiner et al., 2007*; *Iglesias and Devreotes, 2012*; *Stock and Pauli, 2021*). Although the planar-polarized distribution of the WAVE complex across the epithelium was lost in *fat2* mutant tissue, we wondered (1) whether the WAVE complex could still form regions of enrichment in individual cells and (2) whether these WAVE complex-enriched regions were active and responsible for templating unpolarized protrusions. To evaluate the WAVE complex distribution along the edges of individual cells, we generated entirely *fat2* mutant epithelia in which patches of cells expressed Sra1-GFP. At cell-cell interfaces along Sra1-GFP expression boundaries, we found that the boundary cells often had cortical regions devoid of Sra1-GFP (*Figure 6A*). This observation shows that the WAVE complex is not uniformly localized around the cortex and can form regions of enrichment without Fat2. We also saw that Sra1-GFP enrichment coincided with the presence of F-actin protrusions (*Figure 6A*), indicating that the WAVE complex in these regions is active. To confirm that the WAVE complex builds the protrusions in *fat2* epithelia, we co-imaged Sra1-GFP and a membrane label, and found that Sra1-GFP was enriched at the tips of membrane protrusions (*Figure 6—figure supplement 1*; *Figure 6—video 1*). These data indicate that the WAVE complex can still accumulate and build protrusions in the absence of Fat2, tissue-wide planar polarity, and collective cell migration.

A striking feature of migrating follicle cells is the stable polarization of their protrusive leading edges. It is not known whether Fat2 contributes to the stabilization of protrusive regions in addition to positioning them. If so, the positions of WAVE complex-enriched, protrusive regions of *fat2* epithelia should fluctuate more than those of control epithelia, in addition to being less well-polarized at the tissue level. To see if this is the case, we acquired timelapse movies of Sra1-GFP and monitored its distribution along cell perimeters over time. In control epithelia, Sra1-GFP was strongly enriched along leading-trailing interfaces relative to side interfaces over the 20-min timelapse. Side interfaces were mostly devoid of Sra1-GFP, except for infrequent Sra1-GFP accumulations that persisted for several minutes (*Figure 6B–D*; *Figure 6—video 2*). In contrast, in *fat2* epithelia, the regions of greatest Sra1-GFP enrichment along the cell perimeter changed substantially over the 20-min timelapse and multiple Sra1-GFP-enriched regions were often present simultaneously in individual *fat2* cells. Sra1-GFP accumulated in these regions, typically spreading outward along the membrane as it did so, and then dissipated. These events had a duration that was comparable to the transient accumulations of Sra1-GFP at side interfaces in control cells (*Figure 6B–D*; *Figure 6—video 2*). Because all cell-cell interfaces in *fat2* epithelia and side interfaces in control epithelia lack Fat2, this suggests a several-minutes timescale over which regions of WAVE complex enrichment can persist without stabilization by Fat2. Live imaging of Sra1-GFP in *fat2* mosaic epithelia yielded similar information—Sra1-GFP

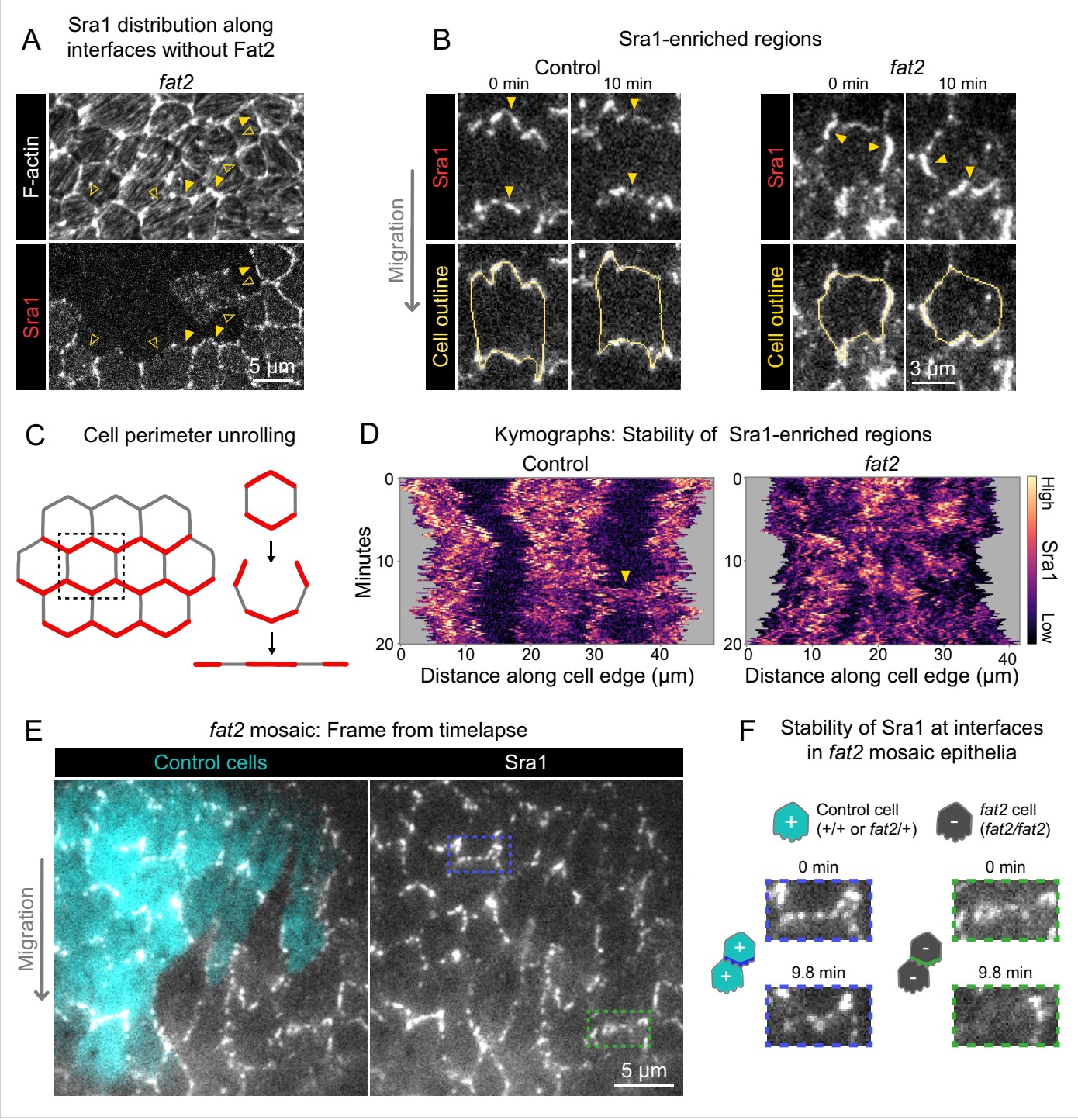

**Figure 6.** Fat2 stabilizes a region of WAVE complex enrichment in each cell. (**A**) Images of phalloidin-stained F-actin and mosaically expressed Sra1-GFP in an entirely *fat2* mutant epithelium. Filled and open arrows indicate genotype boundary interfaces with and without Sra1-GFP enrichment, respectively. Sra1-GFP enrichment is heterogeneous, and interfaces with Sra1-GFP enrichment have more F-actin protrusions. (**B**) Timelapse frames of Sra1-GFP in control and *fat2* epithelia. Top row shows Sra1-GFP with arrows indicating regions of Sra1-GFP enrichment; bottom row shows Sra1-GFP and outlines of cell perimeters used to make kymographs. Laser intensity and brightness display settings differ between genotypes. See related *Figure 6—video 2*. (**C**) Diagram of cell perimeter unrolling for kymograph generation. Red represents planar-polarized Sra1 as distributed before and after unrolling. (**D**) Kymographs of Sra1-GFP fluorescence intensity along cell perimeter outlines exemplified in (**C**). The y-axis length of regions of high Sra1-GFP enrichment reports their stability over time. Control cells have Sra1-GFP regions along leading-trailing interfaces that are stable over 20 minutes. In *fat2* cells, Sra1-GFP-enriched regions are less stable. The arrow indicates a transient accumulation of Sra1-GFP at a control cell side. These occur

*Figure 6 continued on next page*

*Figure 6 continued*

occasionally, and their stability is similar to Sra1-GFP regions in *fat2* cells. (**E**) Timelapse frame of a *fat2* mosaic epithelium in which all cells express Sra1-GFP, used to evaluate Sra1-GFP dynamics in control or *fat2* cells within a migratory context. Boxes indicate a leading-trailing interface between two control cells (blue) or *fat2* cells (green). Representative of 9 similar timelapse movies. See related *Figure 6—video 3*. (**F**) Larger images of the interfaces boxed in (**E**), taken 9.8 min apart. Sra1-GFP is initially enriched along both interfaces. It remains enriched in the control interface throughout, but loses enrichment along the *fat2* interface.

The online version of this article includes the following video and figure supplement(s) for figure 6:

**Figure supplement 1.** The WAVE complex is still associated with protrusions in the absence of Fat2.

**Figure 6—video 1.** Sra1 enrichment at protrusion tips in control and *fat2* epithelia.

https://elifesciences.org/articles/78343/figures#fig6video1

**Figure 6—video 2.** WAVE complex-enriched region dynamics in control and *fat2* epithelia.

https://elifesciences.org/articles/78343/figures#fig6video2

**Figure 6—video 3.** WAVE complex-enriched region dynamics in a fat2 mosaic epithelium.

https://elifesciences.org/articles/78343/figures#fig6video3

enrichment fluctuated more at interfaces between *fat2* cells than interfaces between control cells despite both being in a migratory tissue (*Figure 6E and F*; *Figure 6—video 3*).

To see if Fat2's role stabilizing the WAVE complex distribution translates to a role stabilizing protrusive regions, we returned to our timelapse movies of membrane protrusions in control and *fat2* epithelia, this time focusing on the protrusions' dynamics rather than their distribution. In control cells, protrusion polarity appeared largely stable over the 20-min duration of our timelapse movies, whereas in *fat2* cells it often shifted substantially. In some *fat2* epithelia, protrusion positions shifts were largely restricted to two opposite-facing cell edges, whereas in others, protrusions positions shifted seemingly at random (*Figure 7A*, *Figure 7—video 1*). To evaluate the stability of protrusion polarity quantitatively, we measured the frequency of interface protrusion polarity 'switches', in which first one cell and then its neighbor protruded across their shared interface (*Figure 7B*). These events were rare in control epithelia, with ~2% of interfaces switching polarity per hour. In contrast, they were more common in *fat2* epithelia, with ~60% of interfaces switching polarity per hour (*Figure 7C*). Together, these observations show that, in addition to polarizing the WAVE complex and protrusive activity to the leading edge, Fat2 stabilizes their distributions for repeated cycles of protrusion from one long-lived cell region (*Figure 7D*).

## Fat2 and the WAVE complex colocalize across leading-trailing cell-cell interfaces

Finally, we explored how Fat2 recruits the WAVE complex across the cell-cell interface. To constrain the set of possible mechanisms, we assessed the spatial scale of their interaction. Fat2 has a punctate distribution along each cell's trailing edge (*Viktorinová and Dahmann, 2013*; *Barlan et al., 2017*), so we asked whether Fat2 recruits the WAVE complex locally to these sites, or recruits it more broadly to the entire interface. We evaluated the colocalization between Fat2 and the WAVE complex along leading-trailing interfaces, visualizing Fat2 with an endogenous 3xeGFP tag (Fat2-3xGFP) and the WAVE complex with mCherry-tagged Abi under control of the ubiquitin promoter (Abi-mCherry). Like Fat2-3xGFP, Abi-mCherry formed puncta, and Abi-mCherry and Fat2-3xGFP puncta strongly colocalized (Spearman's $r=0.71 \pm 0.04$; *Figure 8A–E*). Abi-mCherry colocalized significantly less strongly with uniformly-distributed E-cadherin-GFP (Spearman's $r=0.49 \pm 0.07$; *Figure 8—figure supplement 1*,B), indicating that Fat2-3xGFP-Abi-mCherry colocalization was not simply a byproduct of curved membrane morphology or our measurement approach. In timelapse movies, Fat2-3xGFP and Abi-mCherry puncta moved together through cycles of protrusion extension and retraction (*Figure 8B*; *Figure 8—video 1*). Short-lived Abi-mCherry accumulations formed infrequently at cell sides away from Fat2, similar to the Sra1-GFP side accumulations we described earlier (*Figure 6D*; *Figure 8—figure supplement 1*; *Figure 6—video 2*; *Figure 8—video 2*). Together, these findings suggest that Fat2 recruits the WAVE complex locally, at the scale of individual puncta, with the WAVE complex occasionally 'escaping' Fat2-dependent concentration at the leading edge.

If Fat2 puncta locally recruit the WAVE complex, changing the distribution of Fat2 puncta should cause corresponding changes to the distribution of the WAVE complex. To test this, we examined

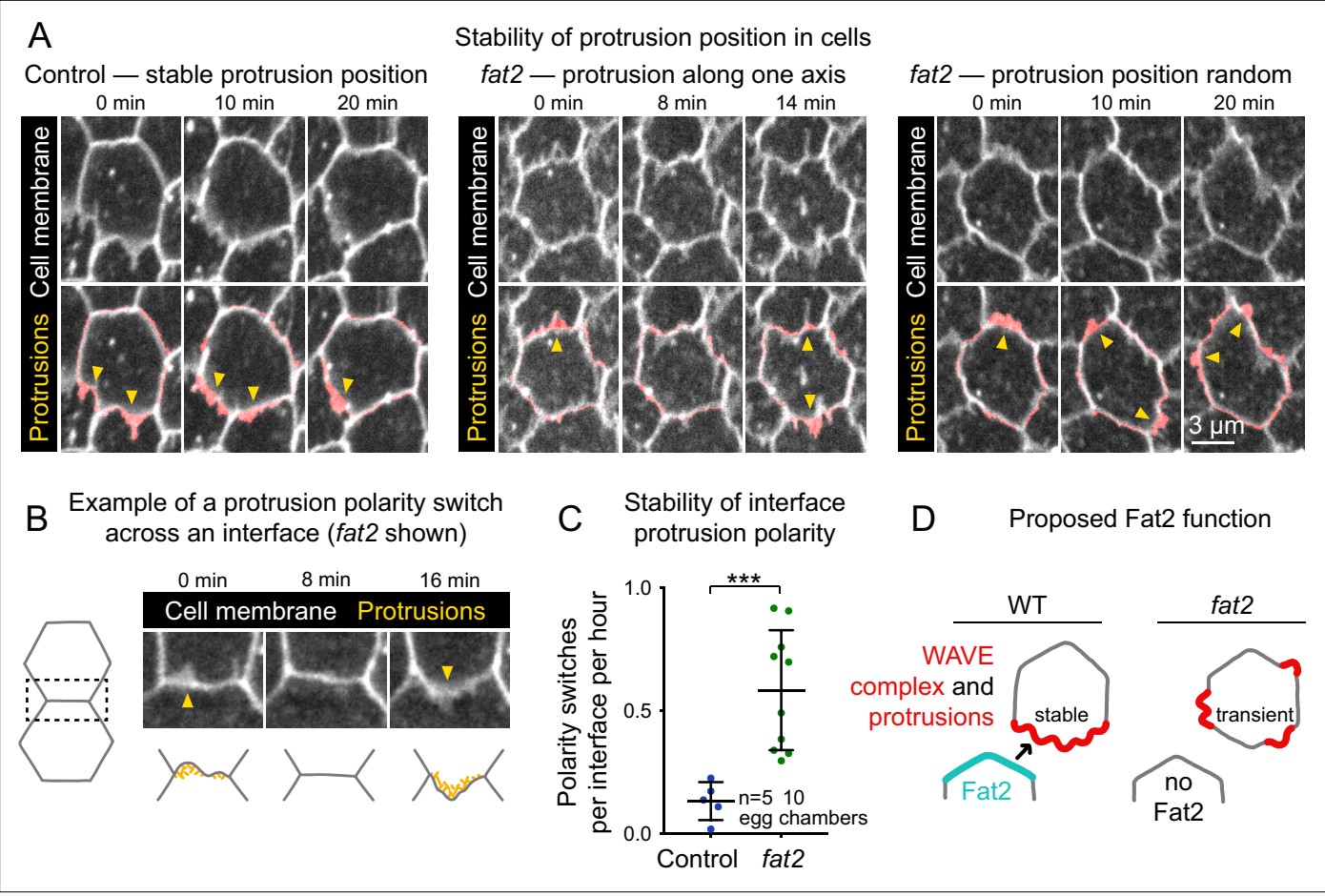

**Figure 7.** Fat2 stabilizes a protrusive region in each cell. (**A**) Timelapse frames of control and *fat2* epithelia labeled with a membrane dye, showing the position of a cell's protrusions over time. Top row shows the interfaces and protrusions of one cell and its neighbors. Segmented membrane extensions originating from the center cell (red) are overlaid in the bottom row. Arrows indicate sites of membrane protrusion. In the control cell, protrusion position is stable, whereas in *fat2* cells it shifts either along a fixed axis (middle) or seemingly at random (right). See related *Figure 7—video 1*. (**B**) Example in which one cell and then its neighbor protrudes across a shared interface (a 'polarity switch'). The row shows timelapse frames of an interface and associated protrusions from a *fat2* epithelium labeled with membrane dye. Arrows originate in the protruding cell and point in the direction of protrusion. The bottom row shows corresponding diagrams of the interface with F-actin-rich protrusions illustrated in yellow. (**C**) Plot showing the frequency of interface protrusion polarity switches (exemplified in **B**) in timelapse movies of control and *fat2* epithelia. Polarity switches occur more frequently at *fat2* interfaces than control ones. Unpaired t-test; ***p=0.0002. Bars indicate mean ± SD. (**D**) Diagram showing the proposed role of Fat2 stabilizing a region of WAVE complex enrichment and protrusivity. Without Fat2, WAVE complex-enriched, protrusive regions are reduced and more transient.

The online version of this article includes the following video and source data for figure 7:

**Source data 1.** Frequency of interface protrusion polarity switches in control and *fat2* epithelia.

**Figure 7—video 1.** Dynamics of protrusive regions in control and *fat2* cells.

https://elifesciences.org/articles/78343/figures#fig7video1

follicle cells expressing an endogenous Fat2 truncation that lacks the intracellular domain (Fat2ΔICD-3xGFP), which distributes more broadly around the cell perimeter than wild-type Fat2 (*Aurich and Dahmann, 2016*; *Barlan et al., 2017*), but remains punctate. The distribution of Abi-mCherry expanded around the cell perimeter in the Fat2ΔICD-3xGFP background (*Figure 8A*) as was previously reported for protrusions (*Barlan et al., 2017*). Despite their altered distributions, Abi-mCherry puncta colocalized just as well with Fat2 ΔICD-3xGFP puncta as with Fat2-3xGFP puncta (Spearman's *r*=0.71 ± 0.04 vs 0.71±0.05; *Figure 8E*; *Figure 8—figure supplement 1*). From these data we conclude that Fat2 controls the distribution of the WAVE complex by concentrating the WAVE complex in adjacent

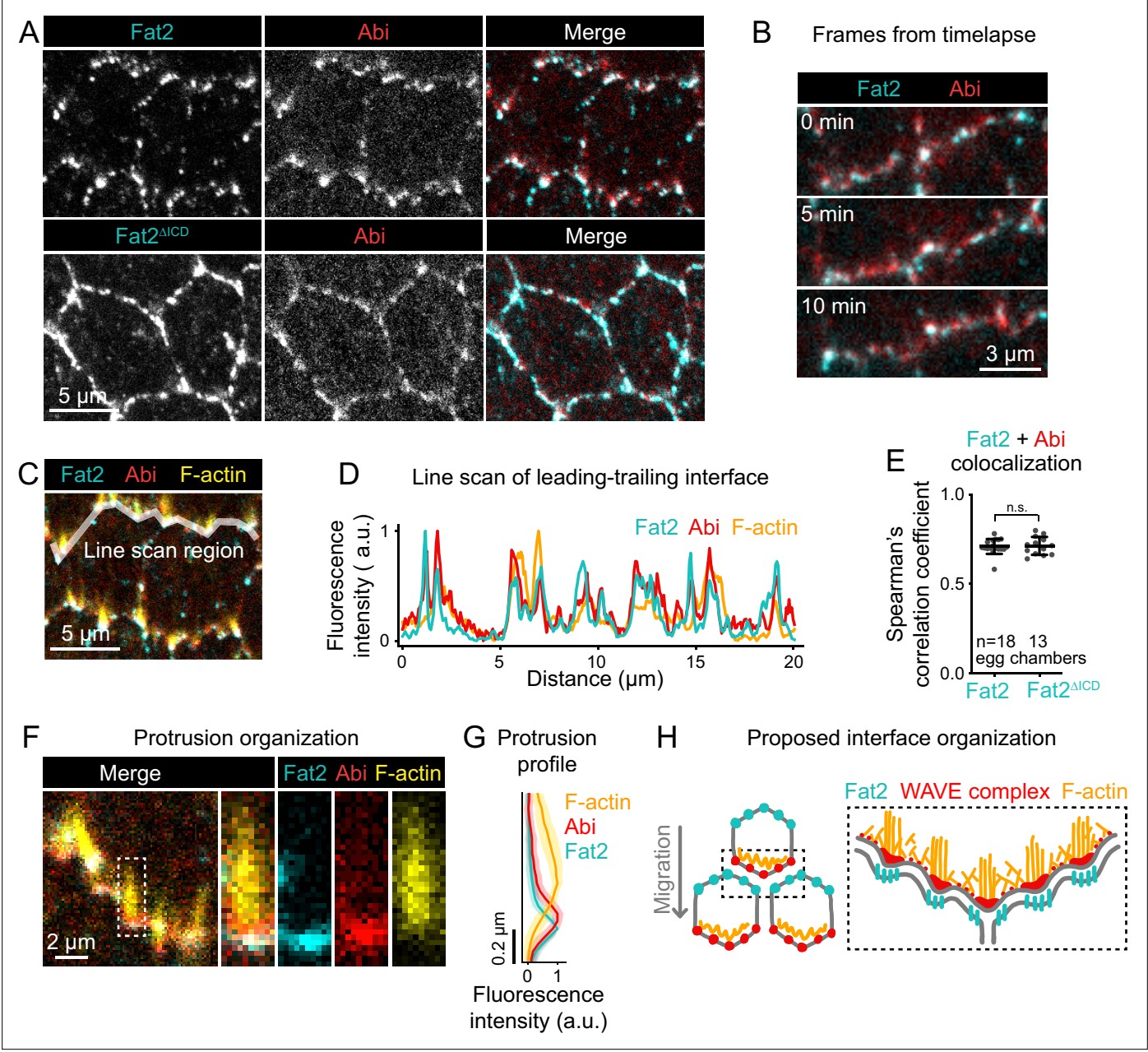

**Figure 8.** Fat2 colocalizes with the WAVE complex across leading-trailing cell-cell interfaces. (**A**) Images of cells expressing Abi-mCherry and endogenous full-length Fat2-3xGFP or endogenous Fat2-3xGFP lacking the intracellular domain (Fat2ΔICD), used to assess colocalization. (**B**) Timelapse frames showing the leading-trailing interfaces of two cells expressing Fat2-3xGFP and Abi-mCherry, showing their colocalization over time. See related *Figure 8—video 1*. (**C**) Image showing the leading-trailing interface region used in (**D**); it is also an example of a region used in (**E**). (**D**) Line scan showing the fluorescence intensity of Fat2-3xGFP, Abi-mCherry, and F-actin (phalloidin) along the leading-trailing interfaces of the two cells in (**C**) showing their corresponding peaks of enrichment. (**E**) Plot of Spearman's correlation coefficients of Fat2-3xGFP or Fat2ΔICD-3xGFP and Abi-mCherry showing no significant difference in colocalization. Bars indicate mean ± SD. One-way ANOVA ($F_{(5,81)}=44.86$, p=0.0164 with *Figure 8—figure supplement 1*) with post-hoc Tukey's test; n.s. p>0.99. (**F**) Image showing the distribution of Fat2-3xGFP, Abi-mCherry, and F-actin (phalloidin) at the leading-trailing interface and along the boxed filopodium. (**G**) Plot showing fluorescence intensity of traces of F-actin, Abi-mCherry, and Fat2-3xGFP showing their relative sites of enrichment along the length of filopodia. Lines and shaded regions indicate mean ± SD. n=74 protrusions (used for SD), 18 cells, 1 cell/egg chamber. (**H**) Diagram of proposed organization of Fat2, the WAVE complex, and F-actin along the leading-trailing interface based on the present data and previously published work (*Viktorinová and Dahmann, 2013*; *Cetera et al., 2014*; *Barlan et al., 2017*). Fat2 puncta at the trailing edge colocalize with WAVE complex puncta at the leading edge, ahead of filopodia embedded within the lamellipodium.

The online version of this article includes the following video, source data, and figure supplement(s) for figure 8:

*Figure 8 continued on next page*

*Figure 8 continued*

**Source data 1.** Colocalization of Fat2 and Abi along leading-trailing interfaces.

**Source data 2.** Fat2, Abi, and F-actin distributions along the length of filopodia.

**Figure supplement 1.** Ena and Lar are not required for colocalization between Fat2 and the WAVE complex.

**Figure supplement 1—source data 1.** Ena, Abi, and F-actin distributions along the length of filipodia.

**Figure supplement 1—source data 2.** Abi enrichment at leading-trailing interfaces in the absence of Lar.

**Figure 8—video 1.** Colocalization of puncta of Fat2 and the WAVE complex along leading-trailing interfaces.
https://elifesciences.org/articles/78343/figures#fig8video1

**Figure 8—video 2.** WAVE complex accumulation at side-facing protrusions away from Fat2.
https://elifesciences.org/articles/78343/figures#fig8video2

puncta. These findings also demonstrate that the Fat2 intracellular domain is dispensable for Fat2-WAVE complex interaction in collectively migrating follicle cells.

Ena-dependent filopodia are embedded within and grow from the lamellipodia (*Cetera et al., 2014*). The WAVE complex interacts with Ena and is required for the filopodia to form (*Cetera et al., 2014*; *Chen et al., 2014b*), so we asked whether the distribution of Fat2-WAVE complex puncta is related to the distribution of filopodia. Labeling filopodia tips with a GFP-tagged Ena transgene (GFP-Ena) and comparing the localization of Abi-mCherry and F-actin to either GFP-Ena or to Fat2-3xGFP, we found that the sites of highest Fat2-3xGFP and Abi-mCherry enrichment coincided with filopodia tips (*Figure 8C, D and F*; *Figure 8—figure supplement 1*). Fluorescence intensity profiles along filopodia lengths showed that Fat2-3xGFP and Abi-mCherry were enriched just ahead of the F-actin-rich region (*Figure 8F and G*). Fat2-3xGFP was shifted slightly forward from Abi-mCherry, consistent with the separation of Fat2-3xGFP and Abi-mCherry fluorophores by a cell-cell interface (*Figure 8F and G*; *Figure 8—figure supplement 1*). This analysis demonstrates a stereotyped organization in which Fat2 and the WAVE complex are concentrated with Ena near the tips of the filopodia, with Fat2 at the trailing edge across the cell-cell interface from the leading edge components (*Figure 8H*).

We considered two explanations for the close spatial relationship between Fat2 puncta, WAVE complex puncta, and filopodia. Fat2 could recruit the WAVE complex locally to puncta, and WAVE complex puncta shape the distribution of filopodia. Alternatively, Fat2 could recruit the WAVE complex to the leading edge, but their colocalization in puncta be a secondary effect of the filopodia, perhaps caused by the known interaction between Ena and Abi (*Chen et al., 2014b*) or by deformation of the leading-trailing interface. To rule out a dependence on filopodia, we measured colocalization between Fat2-3xGFP and Abi-mCherry in *ena*-RNAi-expressing epithelia, in which filopodia are strongly depleted (*Cetera et al., 2014*; *Figure 1F*). Despite the loss of filopodia, both Fat2-3xGFP and Abi-mCherry remained punctate, and their colocalization was only slightly reduced (Spearman's $r$=0.71 ± 0.04 vs 0.65±0.03, *Figure 8—figure supplement 1*,B). We therefore rule out Ena or the filopodia themselves as required mediators of the spatial relationship between Fat2 and the WAVE complex, and infer that Fat2-WAVE complex colocalization is indicative of Fat2 recruitment of the WAVE complex locally to these sites.

Altogether, we propose that Fat2 acts locally, at the scale of individual Fat2 puncta, to concentrate the WAVE complex in adjacent puncta across the cell-cell interface. Because Fat2 puncta are distributed along the trailing edge, this has the broader effect of stabilizing a region of WAVE complex enrichment at the leading edge.

## Discussion

This work demonstrates that a *trans* interaction between the atypical cadherin Fat2 and the WAVE complex can stabilize WAVE complex polarity for directed cell migration. Fat2, localized to the trailing edge of each cell, recruits the WAVE complex to the leading edge of the cell behind, just across their shared interface. By concentrating WAVE complex activity in a restricted region, Fat2 strongly biases lamellipodia and filopodia to form at these leading edge sites, stably polarizing overall cell protrusive activity to one cell side. Because the Fat2-WAVE complex signaling system is deployed at each leading-trailing interface in a planar-polarized manner, it both polarizes protrusions within individual

cells and aligns these individual cell polarities across the epithelium. This allows the cells to exert force in a common direction and achieve a highly coordinated collective cell migration.

While the molecular players differ, local coupling of leading and trailing edges through asymmetric interactions across their shared interface is a recurring motif in studies of epithelial collective cell migrations. In an epithelial cell culture model of collective migration, asymmetric pulling forces across cell-cell interfaces polarize Rac1 activity and cell protrusion (*Das et al., 2015*). In another model, one cell's lamellipodium is stabilized by confinement under the trailing edge of the cell ahead, reinforcing interface asymmetry (*Jain et al., 2020*). In an endothelial collective cell migration model, asymmetric membrane 'fingers' containing VE-cadherin extend from the trailing edge and are engulfed by the leading edge of the cell behind, whose movement they help guide (*Hayer et al., 2016*). These types of leading-trailing edge coupling systems could operate together with longer-range cues to reinforce the planar polarity of cells' migratory structures. In migrations with a closed topology and no extrinsic directional cues, such as that of the follicle cells, local polarity coupling may be especially critical for collective migration.

Our development of new computational tools to segment and quantify membrane protrusion dynamics in a collectively-migrating epithelium has led to new insights into how Fat2 regulates protrusions. We found that without Fat2, protrusivity was reduced, and the distribution of remaining protrusions expanded around the cell periphery. Therefore, Fat2 not only promotes protrusion at the leading edge, but also restricts protrusion to that edge. Analysis of *fat2* mosaic epithelia revealed that Fat2 acts locally to enforce this restriction—even in the context of a globally planar-polarized, migratory epithelium, cells lacking input from Fat2 in the cell ahead were unable to polarize their protrusions in the direction of migration. Although protrusions were no longer biased in one direction without Fat2, the presence of axial orientation bias in a subset of *fat2* epithelia indicates that some form of Fat2-independent planar polarity was still present. This could be mediated by undiscovered planar signaling molecules or by a mechanical cue such as tension transmitted between cells. Investigating the cell-cell communication that gives rise to this layer of planar polarity will be an interesting area for future research.

Excitable WAVE complex activity underlies lamellipodial protrusion (*Weiner et al., 2007*; *Xiong et al., 2010*; *Graziano and Weiner, 2014*), and WAVE complex activity is often entrained by directional cues from the environment (*Millius et al., 2009*; *Xiong et al., 2010*; *Huang et al., 2013*; *Hayashi et al., 2014*). We hypothesize that Fat2 acts as a similar activity-entraining directional cue in follicle cells. Excitable WAVE complex dynamics were especially apparent in our data in contexts where Fat2 was absent from the cell-cell interface—either at interfaces between cells in *fat2* epithelia, or at protrusive side interfaces we observed with low frequency in control cells. In both contexts, the WAVE complex accumulated at an edge region, spread laterally along the membrane, and then dissipated. This corresponded with the initiation, growth, and collapse of a protrusion. Where Fat2 was present, the WAVE complex distribution along the cell perimeter stayed more constant and WAVE complex levels fluctuated in place, but did not appear to spread laterally. These findings, along with the loss of cell and tissue-scale WAVE complex polarization in the absence of Fat2, suggest that Fat2 acts by concentrating WAVE complex activity in a narrow region, thereby polarizing protrusions to a single leading edge.

How might Fat2 locally concentrate the WAVE complex? The WAVE complex is activated by recruitment to the plasma membrane (*Oikawa et al., 2004*; *Lebensohn and Kirschner, 2009*; *Chen et al., 2010*). Positive regulators of WAVE complex accumulation include active Rac, phosphatidylinositol (3,4,5)-triphosphate ($PIP_3$), membrane-localized proteins that directly bind the WAVE complex, and the WAVE complex itself (*Miki et al., 1998*; *Steffen et al., 2004*; *Oikawa et al., 2004*; *Sossey-Alaoui et al., 2005*; *Weiner et al., 2006*; *Nakao et al., 2008*; *Namekata et al., 2010*; *Graziano and Weiner, 2014*; *Chen et al., 2014a*). We propose that Fat2 promotes WAVE complex accumulation within a stable region by acting through one or more of these positive regulators, thereby controlling the site where the WAVE complex excitation threshold is crossed and a protrusion is formed. Under this model, in the absence of Fat2, this site selection instead becomes more stochastic and therefore long-lasting protrusive regions cannot form. If part of the WAVE complex circuit is limiting for protrusion formation, this could also account for Fat2's ability to suppress protrusion formation away from the leading edge. However, there are other possible suppression mechanisms. For example, in neutrophils, protrusions have been shown to increase

membrane tension and thereby suppress distant protrusion, enforcing the selection of a single protrusive region (*Houk et al., 2012*).

Fat2 acts at the trailing edge of each cell to recruit the WAVE complex in trans, so there must be one or more transmembrane proteins at the leading edge of each cell that bridge this interaction. Previous work has shown that the receptor tyrosine phosphatase Lar is part of this bridge—Fat2 recruits Lar to each follicle cell's leading edge (*Barlan et al., 2017*), and in Lar's absence both WAVE complex levels and cell protrusions are reduced (*Barlan et al., 2017*; *Squarr et al., 2016*; *Figure 8—figure supplement 1*). However, the WAVE complex that persists at the leading edges of *lar* cells still colocalizes with Fat2 (*Figure 8—figure supplement 1*). Therefore, there must be at least one other transmembrane protein that works alongside Lar to mediate the Fat2-WAVE complex interaction. Identifying the missing leading edge protein(s) will be important to fully understand how Fat2 shapes WAVE complex activity.

Fat2 is localized in puncta along each cell's trailing edge (*Viktorinová and Dahmann, 2013*; *Barlan et al., 2017*), and we show here that these puncta correspond 1:1 with regions of high WAVE complex enrichment just across the leading-trailing cell-cell interface. Fat2's punctate distribution and its levels along cell-cell interfaces are unaffected by loss of the WAVE complex (*Barlan et al., 2017*), indicating that Fat2 puncta shape the distribution of the WAVE complex and protrusions, not the reverse. We further show that the puncta sit at the tips of filopodia that form within the lamellipodial actin network. Filopodia are a prominent feature of the long-lived protrusive regions that form in wild-type epithelia, but appear to be disproportionately reduced in the short-lived, fluctuating protrusive regions that form in *fat2* epithelia. We therefore propose that by concentrating the WAVE complex and/or stabilizing its distribution, Fat2 also facilitates filopodia formation. It should be noted, however, that the filopodia are dispensable for collective follicle cell migration (*Cetera et al., 2014*), so the reason these structures form remains to be determined.

Why, and how, is Fat2 localized in puncta? Cadherins commonly form puncta, though the causes and functions of this organization vary (*Truong Quang et al., 2013*; *Rubinstein et al., 2017*; *Li et al., 2021*). For example, Flamingo (or mammalian Celsr1), an atypical cadherin and central component of the core planar cell polarity pathway, is stabilized by clustering, and this clustering is important for its planar polarization (*Strutt et al., 2011*; *Cho et al., 2015*; *Stahley et al., 2021*). In future work, it will be important to determine how Fat2 assembles in puncta, and whether this local clustering is important for its polarization to trailing edges or its effect on the organization of leading edges. More broadly, it will be critical to determine how Fat2 achieves its trailing edge localization, a necessary step in the polarization of the tissue.

# Materials and methods

## Key resources table

| Reagent type (species) or resource | Designation | Source or reference | Identifiers | Additional information |
| --- | --- | --- | --- | --- |
| Gene (*Drosophila melanogaster*) | Abi | NA | FLYB:FBgn0020510 | FlyBase Name: Abelson interacting protein |
| Gene (*Drosophila melanogaster*) | Dlg | NA | FLYB:FBgn0001624 | FlyBase Name: discs large 1 |
| Gene (*Drosophila melanogaster*) | E-cadherin | NA | FLYB:FBgn0003391 | FlyBase Name: shotgun |
| Gene (*Drosophila melanogaster*) | Ena | NA | FLYB:FBgn0000578 | FlyBase Name: enabled |
| Gene (*Drosophila melanogaster*) | Fat2 (kug) | NA | FLYB:FBgn0261574 | FlyBase Name: kugelei |
| Gene (*Drosophila melanogaster*) | Lar | NA | FLYB:FBgn0000464 | FlyBase Name: Leukocyte-antigen-related-like |
| Gene (*Drosophila melanogaster*) | Scar | NA | FLYB:FBgn0041781 | FlyBase Name: SCAR |

*Continued on next page*

*Continued*

| Reagent type (species) or resource | Designation | Source or reference | Identifiers | Additional information |
|---|---|---|---|---|
| Gene (*Drosophila melanogaster*) | Sra1 (CYFIP) | NA | FLYB:FBgn0038320 | FlyBase Name: Cytoplasmic FMR1 interacting protein |
| Genetic reagent (*Drosophila melanogaster*) | Abi-mCherry or ubi >Abi-mCherry | Bloomington *Drosophila* Stock Center; FLYB:FBrf0227194 (S. Huelsmann) | FLYB:FBst0058729; BDSC:58729 | FlyBase Symbol: P{Ubi-mCherry.Abi}3 |
| Genetic reagent (*Drosophila melanogaster*) | *abi*-RNAi | National Institute of Genetics, Japan | FLYB:FBtp0079430; NIG:9749 R | |
| Genetic reagent (*Drosophila melanogaster*) | E-cadherin-GFP | Bloomington *Drosophila* Stock Center; PMID:19429710 | FLYB:FBst0060584; BDSC:60584 | FlyBase Genotype: y[1] w*; TI{TI}shg[GFP] |
| Genetic reagent (*Drosophila melanogaster*) | GFP-Ena or ubi >GFP-Ena | Bloomington *Drosophila* Stock Center; FLYB:FBrf0208868 (S. Nowotarski and M. Peiger) | FLYB:FBst0028798; BDSC:28798 | FlyBase Genotype: w*; P{Ubi-GFP.ena}3 |
| Genetic reagent (*Drosophila melanogaster*) | *ena*-RNAi | Vienna *Drosophila* Resource Center | FLYB:FBst0464896; VDRC:43058 | |
| Genetic reagent (*Drosophila melanogaster*) | Fat2-3xGFP FRT80B | Laboratory of S. Horne-Badovinac; PMID:28292425 | FLYB:FBal0326664 | FlyBase Symbol: kug[3xGFP] |
| Genetic reagent (*Drosophila melanogaster*) | Fat2$^{\Delta ICD}$-3xGFP FRT80B | Laboratory of S. Horne-Badovinac; PMID:28292425 | FLYB:FBal0326665 | FlyBase Symbol: kug[ΔICD.3xGFP] |
| Genetic reagent (*Drosophila melanogaster*) | *fat2* or *fat2*$^{N103-2}$ FRT80B | Laboratory of Sally Horne-Badovinac; PMID:22413091 | FLYB:FBal0267777 | FlyBase Symbol: kug[N103-2] |
| Genetic reagent (*Drosophila melanogaster*) | UAS >Flp | Bloomington *Drosophila* Stock Center; PMID:9584125 | FFLYB:FBst0004539; BDSC:4539 | FlyBase Genotype: y[1] w[*]; PUAS-FLP. DJD1 |
| Genetic reagent (*Drosophila melanogaster*) | FRT80B | Bloomington *Drosophila* Stock Center; PMID:8404527 | FLYB:FBti0002073 | FlyBase Symbol: P{neoFRT}80B |
| Genetic reagent (*Drosophila melanogaster*) | UAS >F-Tractin-tdTomato | Bloomington *Drosophila* Stock Center; FLYB:FBrf0226873 (T. Tootle); PMID:24995797 | FLYB:FBst0058989; BDSC:58989 | FlyBase Genotype: w*; P{UASp-F-Tractin. tdTomato}15 A/SM6b; MKRS/TM2 |
| Genetic reagent (*Drosophila melanogaster*) | ubi >GFP NLS (3 L) FRT80B | Bloomington *Drosophila* Stock Center; FLYB:FBrf0108530 (D. Bilder and N. Perrimon) | FLYB:FBst0001620; BDSC:1620 | FlyBase Genotype: w*; P{Ubi-GFP.D}61EF P{neoFRT}80B |
| Genetic reagent (*Drosophila melanogaster*) | *lar*$^{13.2}$ FRT40A | Bloomington *Drosophila* Stock Center; PMID:8598047 | FLYB:FBst0008774; BDSC8774 | |
| Genetic reagent (*Drosophila melanogaster*) | *lar*$^{bola1}$ | Bloomington *Drosophila* Stock Center; PMID:11688569 | FLYB:FBst0091654; BDSC:91654 | |
| Genetic reagent (*Drosophila melanogaster*) | MKRS hsFLP/TM6b, Cre | Bloomington *Drosophila* Stock Center | FLYB:FBst0001501; BDSC:1501 | y[1] w[67c23]; MKRS, P{hsFLP}86E/TM6B, P{Crew}DH2, Tb[1] |

*Continued*

| Reagent type (species) or resource | Designation | Source or reference | Identifiers | Additional information |
|---|---|---|---|---|
| Genetic reagent (*Drosophila melanogaster*) | nanos-Cas9 | Bloomington *Drosophila* Stock Center; FLYB:FBrf0223952 (F. Port and S. Bullock); PMID:25002478 | FLYB:FBst0054591; BSDC:54591 | FlyBase Genotype: y[1] M{nos-Cas9.P} ZH-2A w* |
| Genetic reagent (*Drosophila melanogaster*) | ubi >mRFP NLS (3 L) FRT80B | Bloomington *Drosophila* Stock Center; FLYB:FBrf0210705 (J. Lipsick) | FLYB:FBti0129786; BDSC:30852 | FlyBase Genotype: w1118; P{Ubi-mRFP.nls}3 L P{neoFRT}80B |
| Genetic reagent (*Drosophila melanogaster*) | FRT82b ubi >mRFP NLS (3 R) | Bloomington *Drosophila* Stock Center; FLYB:FBrf0210705 (J. Lipsick) | FLYB:FBst0030555; BDSC:30555 | FlyBase Genotype: w1118; P{neoFRT}82B P{Ubi-mRFP.nls}3 R |
| Genetic reagent (*Drosophila melanogaster*) | Sra1-GFP | Produced for this study | | Sra1 endogenously tagged with GFP using CRISPR. Available from Horne-Badovinac Lab upon request to shorne@uchicago.edu |
| Genetic reagent (*Drosophila melanogaster*) | Sra1-GFP FRT80B | Produced for this study | | Sra1 endogenously tagged with GFP using CRISPR, with FRT80B. Available from Horne-Badovinac Lab upon request to shorne@uchicago.edu |
| Genetic reagent (*Drosophila melanogaster*) | *sra1*-RNAi | Bloomington *Drosophila* Stock Center; PMID:21460824 | FLYB:FBst0038294; BDSC:38294 | FlyBase Genotype: y[1] sc* v[1] sev[21]; P{TRiP.HMS01754}attP2 |
| Genetic reagent (*Drosophila melanogaster*) | tj >Gal4 | National Institute of Genetics, Japan; PMID:12324948 | FLYB:FBtp0089190; DGRC:104055 | FlyBase Symbol: P{tj-GAL4.U} |
| Genetic reagent (*Drosophila melanogaster*) | w[1118] | Bloomington *Drosophila* Stock Center | FLYB:FBal0018186 | |
| Antibody | Discs Large; Dlg (mouse monoclonal) | Developmental Studies Hybridoma Bank | DSHB:4F3; RRID:AB_528203 | (1:20) |
| Antibody | Scar (mouse monoclonal) | Developmental Studies Hybridoma Bank | AB_2618386 | (1:200) |
| Antibody | Alexa Fluor 647, anti-mouse secondary (donkey polyclonal) | Thermo Fisher Scientific | Cat:A31571; RRID:AB_162542 | (1:200) |
| Chemical compound, drug | CellMask Orange Plasma Membrane Stain | Thermo Fisher Scientific | Cat:C10045 | 15 min (1:250) |
| Chemical compound, drug | CellMask Deep Red Plasma Membrane Stain | Thermo Fisher Scientific | Cat:C10046 | 15 min (1:250) |
| Chemical compound, drug | TRITC Phalloidin | Millipore Sigma | Cat:1951 | 15 min at room temp (1:300) |
| Chemical compound, drug | Alexa Fluor 647 phalloidin | Thermo Fisher Scientific | Cat:C10045 | 2 hr at room temp (1:50) |
| Chemical compound, drug | CK-666, Arp2/3 complex inhibitor | Millipore Sigma | Cat:553502 | 750 µM |
| Chemical compound, drug | Formaldehyde, 16%, methanol free, ultra pure | Polysciences | Cat:18814–10 | |
| Chemical compound, drug | Recombinant human insulin | Millipore Sigma | Cat:12643 | |

*Continued on next page*

*Continued*

| Reagent type (species) or resource | Designation | Source or reference | Identifiers | Additional information |
|---|---|---|---|---|
| Recombinant DNA reagent | plasmid: pU6-BbsI-chiRNA | Addgene | Addgene:45946; RRID:Addgene_45946 | PMID:23709638 |
| Recombinant DNA reagent | plasmid: pU6 chiRNA Sra1 C-term | Produced for this study | | CRISPR chiRNA construct for generation of Sra1-GFP. available from Horne-Badovinac Lab upon request to shorne@uchicago.edu |
| Recombinant DNA reagent | plasmid: pDsRed-attP | Addgene | Addgene:51019; RRID:Addgene_51019 | PMID:24478335. Vector used to make pDsRed-attP Sra1-GFP HR |
| Recombinant DNA reagent | plasmid: pTWG | *Drosophila* Genome Resource Center | DGRC:1076 | source of enhanced GFP for generation of Sra1-GFP |
| Recombinant DNA reagent | plasmid: pDsRed-attP Sra1-GFP HR | Produced for this study | | CRISPR homologous recombinaton construct for generation of Sra1-GFP. Available from Horne-Badovinac Lab upon request to shorne@uchicago.edu |
| Software, algorithm | Zen Blue | Zeiss | | |
| Software, algorithm | MetaMorph | Molecular Devices | | |
| Software, algorithm | FIJI (ImageJ) | PMID:22743772 | | |
| Software, algorithm | GraphPad Prism 9 for Mac | GraphPad Software | | |
| Software, algorithm | Microsoft Excel for Mac, version 16.47 | Microsoft | | |
| Software, algorithm | Python 3 | Python Software Foundation | | |
| Software, algorithm | imageio | imageio contributors | | |
| Software, algorithm | matplotlib | The Matplotlib Development team | | |
| Software, algorithm | napari | napari contributors | | |
| Software, algorithm | numpy | numpy contributors | | |
| Software, algorithm | pims | pims contributors | | |
| Software, algorithm | pandas | pandas contributors | | |
| Software, algorithm | scikit-image | scikit-image development team | | |
| Software, algorithm | scikit-ffm | scikit-fmm contributors | | |
| Software, algorithm | scipy | scipy contributors | | |

## Materials, data, and code availability

The code necessary to reproduce core aspects of data analysis, along with numerical data not included in source data files, are available at https://github.com/a9w/Fat2_polarizes_WAVE (*Williams and Donoughe, 2022*). Sequences of plasmids generated in this study are also available at https://github.com/a9w/Fat2_polarizes_WAVE (copy archived at swh:1:rev:0e1ee58588365bd3fba0099c-6f002993a18ec279, *Williams, 2022*). We will share the flies or plasmids themselves upon request to the corresponding author. Image and movie data are available from https://doi.org/10.6084/m9.figshare.20759314.v1.

## *Drosophila* sources, care, and genetics

The sources and references of all stocks used in this study are listed in Key resources table and the genotypes of *Drosophila* used in each experiment and associated figure panels are listed in *Table 1*. *Drosophila* were raised at 25 °C and fed cornmeal molasses agar food. Females 0–3 days post-eclosion were aged on yeast with males prior to dissection. In most cases, they were aged for 2–3 days at 25 °C. Temperatures and yeasting times used for each experiment are reported in *Table 2*. In all RNAi

**Table 1.** Experimental genotypes.

| Figure | Panel | Name | Genotype |
|---|---|---|---|
| 1 | D | F-actin +Ena + Abi | $w$; ubi >GFP-Ena$^{BDSC:28798}$/ubi >Abi-mCherry$^{BDSC:58729}$ |
| 1 | F | Control | $w$; tj >Gal4$^{DGRC:104055}$/+ |
| 1 | F | *ena*-RNAi | $w$; tj >Gal4$^{DGRC:104055}$/UAS-*ena*-RNAi$^{VDRC:43058}$ |
| 1 | F | *abi*-RNAi | $w$; tj >Gal4$^{DGRC:104055}$/+; UAS-*abi*-RNAi$^{NIG:9749R-3}$ |
| 2 | Top row | Protrusion in 1 direction | $w^{1118}$ |
| 2 | Bottom row | Protrusion in both directions | $w$; *fat2*$^{N103-2}$ FRT80B |
| 3 | A-C | Control | $w^{1118}$ |
| 3 | A-C | *fat2* | $w$; *fat2*$^{N103-2}$ FRT80B |
| 3 | A-C | CK-666 | $w^{1118}$ |
| 3 | D | Control | $w^{1118}$ |
| 3 | D | *fat2* | $w$; *fat2*$^{N103-2}$ FRT80B |
| 4 | A,B | *fat2* mosaic | $w$; tj >Gal4$^{DGRC:104055}$, UAS >Flp$^{BDSC:4539}$/+; *fat2*$^{N103-2}$ FRT80B/ubi >GFP NLS FRT80B$^{BDSC1620}$ |
| 5 | B | Sra1-GFP mosaic | $w$; tj >Gal4$^{DGRC:104055}$, UAS >Flp$^{BDSC:4539}$/+; FRT82B Sra1-GFP/FRT82B ubi >mRFP-NLS$^{BDSC:30555}$ |
| 5 | C | *fat2* mosaic | $w$; tj >Gal4$^{DGRC:104055}$, UAS >Flp$^{BDSC:4539}$/+; *fat2*$^{N103-2}$ FRT80B/ubi >GFP NLS FRT80B$^{BDSC1620}$ |
| 5 | D-F | Sra1-GFP mosaic | $w$; tj >Gal4DGRC:104055, UAS >Flp$^{BDSC:4539}$/+; *fat2*$^{N103-2}$ FRT80B Sra1-GFP/ubi >mRFP NLS FRT80$^{BDSC:30852}$ |
| 6 | A | Sra1-GFP mosaic +*fat2* | $w$; tj >Gal4$^{DGRC:104055}$, UAS >Flp$^{BDSC:4539}$/+; *fat2*$^{N103-2}$ FRT80B Sra1-GFP/*fat2*$^{N103-2}$ FRT80B FRT82B |
| 6 | B,D | Control | $w$;; Sra1-GFP/+ |
| 6 | B,D | *fat2* | $w$;; *fat2*$^{N103-2}$ FRT80B Sra1-GFP/*fat2*$^{N103-2}$ FRT80B |
| 6 | E,F | *fat2* mosaic +Sra1 | $w$; tj >Gal4$^{DGRC:104055}$, UAS >Flp$^{BDSC:4539}$/+; *fat2*$^{N103-2}$ FRT80B Sra1-GFP/ubi >mRFP NLS FRT80$^{BDSC:30852}$ |
| 7 | A,C | Control | $w^{1118}$ |
| 7 | A,C | *fat2* | $w$;; *fat2*$^{N103-2}$ FRT80B |
| 7 | B | Example of switch | $w$;; *fat2*$^{N103-2}$ FRT80B |
| 8 | A,E | Fat2 +Abi | $w$;; ubi >Abi-mCherry$^{BDSC:58729}$, Fat2-3xGFP FRT80B/Fat2-3xGFP FRT80B |
| 8 | A,E | Fat2ΔICD + Abi | $w$;; ubi >Abi-mCherry$^{BDSC:58729}$, Fat2$^{ΔICD}$-3xGFP FRT80B/Fat2$^{ΔICD}$-3xGFP FRT80B |
| 8 | B | Fat2 +Abi | $w$;; ubi >Abi-mCherry$^{BDSC:58729}$, Fat2-3xGFP FRT80B/Fat2-3xGFP FRT80B |
| 8 | C,D,F,G | Fat2 +Abi + F-actin | $w$;; ubi >Abi-mCherry$^{BDSC:58729}$, Fat2-3xGFP FRT80B/Fat2-3xGFP FRT80B |
| 3S1 | A | Control | $w^{1118}$ |
| 3S1 | A | *fat2* | $w$;; *fat2*$^{N103-2}$ FRT80B |
| 3S1 | A | CK-666 | $w^{1118}$ |
| 3S1 | B | Control | $w^{1118}$ |
| 3S1 | B | *fat2* | $w$;; *fat2*$^{N103-2}$ FRT80B |
| 3S2 | A-C | Control | $w$; tj >Gal4$^{DGRC:104055}$/+ |
| 3S2 | A-C | *fat2* | $w$; tj >Gal4$^{DGRC:104055}$/+; *fat2*$^{N103-2}$ FRT80B |
| 3S2 | A-C | *abi*-RNAi | $w$; tj >Gal4$^{DGRC:104055}$/+UAS-*abi*-RNAiNIG:9749R-3/+ |
| 3S2 | D | Control | $w$; tj >Gal4$^{DGRC:104055}$/UAS >F-Tractin-tdTomato$^{BDSC:58989}$ |
| 3S2 | D | *fat2* | $w$; tj >Gal4$^{DGRC:104055}$/UAS >F-Tractin-tdTomato$^{BDSC:58989}$; *fat2*$^{N103-2}$ FRT80B |
| 3S2 | D | *abi*-RNAi | $w$; tj >Gal4$^{DGRC:104055}$/UAS >F-Tractin-tdTomato$^{BDSC:58989}$; UAS-*abi*-RNAi$^{NIG:9749R-3}$/+ |
| 3S2 | E,F | Control | $w^{1118}$ |

*Table 1 continued on next page*

*Table 1 continued*

| Figure | Panel | Name | Genotype |
|---|---|---|---|
| 3S2 | E,F | *fat2* | *w;; fat2*$^{N103-2}$ FRT80B |
| 5S1 | A | Sra1-GFP | *w;;* Sra1-GFP |
| 5S1 | A | anti-SCAR | *w;;* Sra1-GFP |
| 5S1 | A | ubi >Abi-mCherry | *w;;* ubi >Abi-mCherry$^{BDSC:58729}$/+ |
| 5S1 | B | Control | *w;* tj >Gal4$^{DGRC:104055}$/+ |
| 5S1 | B | *abi*-RNAi | tj >Gal4$^{DGRC:104055}$/+; UAS-*abi*-RNAi$^{NIG:9749R-3}$/+ |
| 5S1 | C | Control | *w*$^{1118}$ |
| 5S1 | C | Sra1-GFP x1 | *w;;* Sra1-GFP/+ |
| 5S1 | C | Sra1-GFP x2 | *w;;* Sra1-GFP |
| 5S1 | C | *sra1*-RNAi | *w;* tj >Gal4$^{DGRC:104055}$/+; UAS >sra1-RNAi$^{BDSC:38294}$/+ |
| 5S1 | D | Control | *w*$^{1118}$ |
| 5S1 | D | Sra1-GFP x1 | *w;;* Sra1-GFP/+ |
| 5S1 | D | Sra1-GFP x2 | *w;;* Sra1-GFP |
| 5S2 | A, C-E | Control | *w;;* Sra1-GFP/+ |
| 5S2 | A, C-E | *fat2* | *w;; fat2*$^{N103-2}$ FRT80B Sra1-GFP/*fat2*$^{N103-2}$ FRT80B |
| 6S1 | | Control | *w;;* Sra1-GFP/+ |
| 6S1 | | *fat2* | *w;; fat2*$^{N103-2}$ FRT80B Sra1-GFP/*fat2*$^{N103-2}$ FRT80B |
| 8S1 | A,B | Control Fat2 +Abi | *w;;* ubi >Abi-mCherry$^{BDSC:58729}$, Fat2-3xGFP FRT80B/Fat2-3xGFP FRT80B |
| 8S1 | A,B | Fat2ΔICD + Abi | *w;;* ubi >Abi-mCherry$^{BDSC:58729}$, Fat2$^{ΔICD}$-3xGFP FRT80B/Fat2$^{ΔICD}$-3xGFP FRT80B |
| 8S1 | A,B | *ena*-RNAi, Fat2 +Abi | *w;* tj >Gal4$^{DGRC:104055}$/UAS >ena RNAi$^{VDRC:43058}$; ubi >Abi-mCherry$^{BDSC:58729}$, Fat2-3xGFP FRT80B/Fat2-3xGFP FRT80B |
| 8S1 | A,B | Control Fat2 +Abi | *w;* lar$^{bola\ 1BDSC:91654}$/lar$^{13.2\ BDSC:8774}$ FRT40A; ubi >Abi-mCherry$^{BDSC:58729}$, Fat2-3xGFP FRT80B/Fat2-3xGFP FRT80B |
| 8S1 | C | Fat2 +Abi | *w;;* ubi >Abi-mCherry$^{BDSC:58729}$, Fat2-3xGFP FRT80B/Fat2-3xGFP FRT80B |
| 8S1 | D-F | Ena +Abi + F-actin | *w;* ubi >GFP-Ena$^{BDSC:28798}$/ubi >Abi-mCherry$^{BDSC:58729}$ |
| 8S1 | G | Control | *w;;* ubi >Abi-mCherry$^{BDSC:58729}$, Fat2-3xGFP FRT80B/Fat2-3xGFP FRT80B |
| 8S1 | G | *lar* | *w;* lar$^{bola\ 1BDSC:91654}$/lar$^{13.2\ BDSC8774}$ FRT40A; ubi >Abi-mCherry$^{BDSC:58729}$, Fat2-3xGFP FRT80B/Fat2-3xGFP FRT80B |

experiments, *traffic jam* >Gal4 (*tj* >Gal4) (*Hayashi et al., 2002*) was used to drive RNAi expression in follicle cells and not in germ cells. Sra1-GFP and *fat2* mosaic epithelia were generated using the Flp/FRT method (*Golic and Lindquist, 1989*; *Golic, 1991*), using FRT82B and FRT80B recombination sites, respectively. In both cases, *tj* >Gal4 was used to drive expression of UAS >Flp recombinase.

## Generation of Sra1-GFP

Endogenous Sra1 was tagged C-terminally with enhanced GFP (GFP) following the general approaches described by *Gratz et al., 2013* and *Gratz et al., 2014*. The guide RNA target sequence 5'-<u>GCTTAAAT GCATCCCTTTCC</u>**GGG**-3' was chosen with flyCRISPR Target Finder (*Gratz et al., 2014*). The underlined sequence was cloned into the pU6-BbsI-chiRNA plasmid, and the bold sequence is the adjacent PAM motif. For homologous recombination, homology arms approximately 2 kb long flanking the insertion target site were amplified from genomic DNA from the y1 M{nos-Cas9.P}ZH-2A w* (nanos >Cas9) (*Port et al., 2014*) background. GFP was amplified from the pTWG plasmid. A linker with sequence encoding the amino acids 'GSGGSGGS' was added to the N-terminal side of GFP. Homology arms, linker, and GFP were inserted into donor plasmid pDsRed-attP, which contains 3xP3 >DsRed

**Table 2.** Yeasting conditions.

| Figure | Panel | Days on yeast | Temp. (°C) |
|---|---|---|---|
| 1 | D | 2–3 | 25 |
| 1 | F | 3 | 29 |
| 2 | | 2–3 | 25 |
| 3 | A-D | 2–3 | 25 |
| 4 | A,B | 2–3 | 25 |
| 5 | B | 7 | 25 |
| 5 | C | 3 | 25 |
| 5 | D-F | 3 | 25 |
| 6 | A | 5 | 25 |
| 6 | B,D | 2–3 | 25 |
| 6 | E | 2–3 | 25 |
| 7 | A-C | 2–3 | 25 |
| 8 | A,E | 2–3 | 25 |
| 8 | B | 2–3 | 25 |
| 8 | C,D,F,G | 2–3 | 25 |
| S1 | A-C | 2–3 | 25 |
| S2 | A-C,E,F | 2–3 | 29 |
| S2 | D | 2–3 | 29 |
| S3 | A | 2–3 | 25 |
| S3 | B | 3 | 29 |
| S3 | C | 3 | 29 |
| S3 | D | 2–3 | 25 |
| S4 | A-E | 2–3 | 25 |
| S5 | | 2–3 | 25 |
| S6 | A,B,F | 3 | 29 |
| S6 | C | 2–3 | 25 |
| S6 | D-F | 2–3 | 25 |

flanked by loxP sites for insertion screening and subsequent removal. The linker-GFP insertion was made immediately before the Sra1 stop codon. Guide and homologous recombination plasmids were injected by Genetivision Inc into the nanos >Cas9 background. F1 males were screened for 3xP3 >DsRed and then 3xP3 >DsRed was excised by crossing to Cre-expressing flies (MKRS hsFLP/ TM6b Cre).

## Egg chamber dissection

All data come from stage 6–7 egg chambers. To obtain these, ovaries were dissected into live imaging media (Schneider's *Drosophila* medium with 15% fetal bovine serum and 200 µg/mL insulin) in a spot plate using 1 set of Dumont #55 forceps and 1 set of Dumont #5 forceps. Ovarioles were removed from the ovary and from ovariole muscle sheaths with forceps. For live imaging, egg chambers older than the egg chamber to be imaged were removed from the ovariole strands by cutting through the stalk with a 27-gauge hypodermic needle. For fixed imaging, egg chambers older than stage 9 were removed prior to fixation. Removal of older egg chambers allows more compression of the imaged egg chamber between the slide and coverslip so that the basal surface of a field of cells can be imaged in a single plane. For a more detailed description and movies of dissection, see *Cetera et al., 2016* .

## Live imaging sample preparation

Following dissection, ovarioles were transferred to a fresh well of live imaging media. For membrane staining, CellMask Orange or Deep Red plasma membrane stain (Thermo Fisher Scientific, Waltham, MA, 1:500) was added and ovarioles incubated for 15 min, followed by a wash in live imaging media to remove excess stain before mounting. Ovarioles were then transferred to a glass slide with 20 µL of live imaging media. For CK-666 treatment, following plasma membrane staining, ovarioles were transferred to live imaging media with 750 µM CK-666 (Millipore Sigma, St. Louis, MO) and then mounted in the same media. Glass beads with diameter 51 µm were added to support the 22x22 mm #1.5 coverslip and limit egg chamber compression. Coverslip edges were sealed with melted petroleum jelly to prevent evaporation while imaging. Samples were checked for damage using the membrane stain or other fluorescent markers as indicators, and excluded if damage was observed. Slides were used for no more than 1 hr.

## Immunostaining and F-actin staining

Following dissection, ovarioles were fixed in 4% EM-grade formaldehyde in PBT (phosphate buffered saline +0.1% Triton X-100) and then washed 3x5 min in PBT at room temperature. Egg chambers were incubated with primary antibodies in PBT overnight at 4° C (anti-Scar, 1:200) or for 2 hr at room temperature (anti-Discs Large, 1:20) while rocking. Ovarioles were then washed 3x5 min in PBT and

incubated in secondary antibody diluted 1:200 in PBT for 2 hr at room temperature while rocking. F-actin staining was performed using either TRITC phalloidin (Millipore Sigma, 1:250) or Alexa Fluor 647 phalloidin (Thermo Fisher Scientific, 1:50). If TRITC phalloidin was the only stain or antibody used, it was added directly to the fixation media for 15 min of staining concurrent with fixation. Otherwise, TRITC phalloidin was added for 15–30 min at room temperature as the final staining step. Alexa Fluor 647 phalloidin staining was performed for 2 hr at room temperature while the sample was rocking, concurrent with secondary antibody staining where applicable. Ovarioles were then washed 3x5 min in PBT and mounted in 40 μL SlowFade Diamond antifade on a slide using a 22x50 mm #1.5 coverslip, sealed with nail polish, and stored at 4° C until imaged.

## Microscopy

### Laser scanning confocal microscopy

Laser scanning confocal microscopy was used for all fixed imaging and for live imaging of membrane-dyed egg chambers. Imaging was performed with a Zeiss LSM 800 upright laser scanning confocal with a 40 x/1.3 NA EC Plan-NEOFLUAR oil immersion objective or a 63 x/1.4 NA Plan-APOCHROMAT oil immersion objective, diode lasers (405, 488, 561, and 640 nm), and GaAsP detectors. The system was controlled with Zen 2.3 Blue acquisition software (Zeiss). Imaging was performed at room temperature. All images show the basal surface of stage 6–7 egg chambers except for *Figure 5—figure supplement 1A*, bottom row, which shows follicle cells in cross-section. Cross-section images were used for egg chamber staging throughout. Laser scanning confocal microscopy was used to acquire the data in *Figure 1D and F*; *Figure 2*; *Figure 3*; *Figure 3—figure supplement 1A-C*; *Figure 3—figure supplement 2*; *Figure 3—video 1*; *Figure 3—video 3*; *Figure 4*; *Figure 4—video 1*; *Figure 5B–F*; *Figure 5—figure supplement 1*; *Figure 5—figure supplement 2*; *Figure 6A*; *Figure 6—figure supplement 1*; *Figure 6—video 1*; *Figure 7A–C*; *Figure 7—video 1*; *Figure 8A and C–G*; *Figure 8—figure supplement 1*.

### TIRF microscopy

Near-TIRF microscopy was used to visualize Fat2-GFP, Sra1-GFP, Abi-mCherry, and F-Tractin-tdTomato (*Spracklen et al., 2014*) dynamics at the basal surface. Near-TIRF imaging was performed with a Nikon ECLIPSE-Ti inverted microscope with Ti-ND6-PFS Perfect Focus Unit, solid-state 50 mW 481 and 561 nm Sapphire lasers (Coherent technology), motorized TIRF illuminator, laser merge module (Spectral Applied Research), Nikon CFI 100 x Apo 1.45 NA oil immersion TIRF objective with 1.5 x intermediate magnification, and Andor iXon3 897 electron-multiplying charged-coupled device (EM-CCD) camera. Image acquisition was controlled using MetaMorph software. For two color imaging, frames were collected for each color consecutively with the TIRF illumination angle adjusted in between. Imaging was performed at room temperature. For display, movies were corrected for bleaching using the histogram matching method in Fiji (ImageJ) (*Schindelin et al., 2012*; *Schindelin et al., 2015*). TIRF microscopy was used to acquire the data in *Figure 3—figure supplement 2*; *Figure 3—video 2*; *Figure 6B and D–F*; *Figure 6—video 2*; *Figure 6—video 3*; *Figure 8B*; *Figure 8—figure supplement 1*; *Figure 8—video 1*; *Figure 8—video 2*.

## Cell and protrusion segmentation from timelapses of cell membrane

Protrusions from timelapse datasets of the follicle cell basal surface stained with CellMask Orange (see Live imaging sample preparation) were segmented with the Python scikit-image and scipy libraries (*Figure 2*; *van der Walt et al., 2014*; *Virtanen et al., 2020*). First, each cell was segmented and tracked, with manual corrections to cell-cell interface placements made using napari (*napari contributors, 2019* ). Next, a watershed-based approach was used to segment the regions of high fluorescence intensity at the interface of each pair of neighboring cells. This segmented shape encompasses the cell-cell interface and any associated protrusions from either neighboring cell. Last, to assign protrusions to the cell from which they originated, the segmented region was divided in two by the shortest path between its bounding vertices that lay entirely within the region. This approximates the position of the interface between the cells, and in subsequent steps we will call this line 'the interface'. Each of the two resulting protrusion shapes was assigned as originating from the cell on the opposite side of the interface, because protrusions extend from one cell and overlap the other. Using this

approach, all of the protrusive structures that emerge from one cell, and that overlap a single neighboring cell, are grouped together as a single segmented region for subsequent analysis.

## Measurement of membrane protrusivity, protrusion length, and protrusion orientation

After cell edges and associated protrusions were segmented, they were categorized as either protrusive or non-protrusive and their lengths and orientations using Python scikit-fmm, scikit-image, and scipy libraries. We use the term 'membrane extensions' to refer to the cell edge shapes before the protrusive ones have been identified. To measure the length of a membrane extension, we used two different metrics, each of which provides a single length value per cell edge. In one, we calculated the 'average length' of a membrane extension as the membrane extension's area divided by the length of the interface it extended across. As an alternate length measurement, we calculated its 'longest length'. To do so, we first found its 'tip', defined as the pixel within the segmented region farthest from any point along the interface. We then found its 'base', the pixel along the interface that was closest to the tip. We defined membrane extension longest length as the length of the shortest path between base and tip that lay entirely within the membrane extension. To categorize membrane extensions as *protrusive* or *non-protrusive* throughout the study, we used the 'average length' metric. We measured the average length distribution in CK-666-treated epithelia, which are nearly non-protrusive and so provided a measure of the width of the cell-cell interface alone. For all conditions, we categorized a membrane extension as protrusive if its average length was greater than the 98th percentile of length of CK-666-treated epithelia. We then defined the protrusivity of an entire epithelium as the ratio of protrusive to total cell edges in the field of view. We also report two alternate measurements of the protrusivity of an epithelium. In one, We calculate epithelial protrusivity as above, but substitute the longest length as our length measurement (*Figure 3—figure supplement 1A*). In a second, cutoff-independent epithelial protrusivity measurement, we report the epithelium-mean average membrane extension length (*Figure 3—figure supplement 1B*). Swarm plots of each of these analyses were generated using GraphPad Prism 9 (GraphPad, San Diego, CA), as were all other swarm plots.

For analysis of protrusion orientation, we included only the membrane extensions categorized as protrusive according to the 'average length' metric. We defined a protrusion's orientation as the orientation of the vector from its base to its tip. Polar histograms, generated in Python with matplotlib (*Hunter, 2007*), show the distribution of protrusion orientations. In these plots, bar area is proportional to the number of protrusions in the corresponding bin. We note that cells of migratory epithelia often have rearward-pointing retraction fibers as well as protrusions, and our protrusion segmentation method does not distinguish these two types of membrane extensions. For this reason, the degree of protrusion polarity we measure for the migratory control epithelia is likely an underestimate.

## Quantification of F-actin and Sra1-GFP cell-cell interface and non-interface basal surface fluorescence

Cells and cell-cell interfaces were segmented as described above. Cells and interfaces in contact with the tissue border or image border were excluded from analysis. For interface fluorescence intensity, interfaces were dilated by 5 pixels, and mean fluorescence intensity calculated from within this region. Non-interface basal surface fluorescence intensity was calculated as the mean of the remaining (non-interface) tissue surface. For F-actin cell-cell interface enrichment measurements, the overall brightness of the phalloidin staining varied between epithelia independent of genotype. To control for this variation we subtracted the mean intensity of the epithelium's non-interface basal surface from its mean interface intensity measurement. This value, the degree of F-actin interface enrichment, was used as a proxy for F-actin protrusivity.

## Quantification of F-actin and Sra1-GFP planar polarity

As a simple planar polarity measurement, we quantified mean F-actin (phalloidin) or Sra1-GFP fluorescence intensity along each cell-cell interface as a function of the interface's orientation with respect to the anterior-posterior axis. To do this, cells and cell-cell interfaces were segmented as described above. For interface angle measurements, the angular distance between the line defined by the interface-bounding vertices and the anterior-posterior (horizontal) axis was calculated. For interface fluorescence intensity measurements, interface regions were identified as segmented interfaces

dilated by 5 pixels. Vertices, dilated by 10 pixels, were excluded from interface regions. Mean fluorescence intensity was calculated within each interface region, and background (the mean non-interface basal surface fluorescence intensity of all cells in the image) was subtracted. Polar bar plots, generated in Python with matplotlib, show the mean interface intensity as a function of interface angle. In these plots, bar area is proportional to intensity, and Control and *fat2* datasets are rescaled separately so that each have a mean value of one. As a summary statistic for the degree of planar polarization of F-actin or Sra1-GFP in each egg chamber, we found the average fluorescence intensities of interfaces with angles between 0° and 10° and between 80° and 90°. These correspond with leading-trailing and side interfaces, respectively, in migratory epithelia. The *leading-trailing interface enrichment* is the ratio of these numbers.

## Autonomy analysis in mosaic epithelia

Egg chambers were stained with Alexa Fluor 647 phalloidin to mark protrusions, which indicate migration direction, and to determine whether egg chambers were planar-polarized. We analyzed only S6-7 egg chambers with mixtures of control and *fat2* cells that had global stress fiber alignment orthogonal to the anterior-posterior axis, indicating global planar polarity. Since migration is required to maintain planar polarity (*Cetera et al., 2014*), this also indicates that the epithelium was migratory. We then measured the Sra1-GFP fluorescence intensity at leading-trailing interfaces and medial basal surfaces to determine whether changes in Sra1-GFP levels coincided with the genotype of the Sra1-containing cell, or the genotype of the cell ahead. To select leading-trailing interface regions to measure, we drew 10 pixel-wide segmented lines along leading edges of individual cells, assigning them to a condition based on their own genotype and the genotype of the cell just ahead. If a cell had both control and *fat2* cells ahead of it, we measured those leading edge segments separately, assigning each to the applicable condition. Lines were drawn along all visible, in-focus *fat2*-control and control-*fat2* boundaries and a similar number of control-control and *fat2*-*fat2* boundaries. Epithelia were excluded if two or more interfaces of each of the four genotype combinations were not present. From these regions, we measured mean Sra1-GFP fluorescence intensity for each cell, and then took the mean of these as the fluorescence intensity per egg chamber. For a diagram of this method, see *Barlan et al., 2017*, which we have modified here to allow measurement of individual cells. To quantify medial basal surface Sra1-GFP fluorescence intensity, we used the same approach with the following exceptions: we measured polygonal regions of the basal surface of individual cells away from cell-cell interfaces, and cells were excluded if their leading edge contacted both control and *fat2* cells.

## Quantification of migration rate

Egg chambers were dissected, dyed with CellMask Orange, and mounted for live imaging as described above. Several ovarioles were mounted on each slide, with each ovariole terminating in a S6-7 egg chamber. Timelapse imaging was performed for 30 min with frames acquired every 30 s. Multi-point acquisition was used to obtain movies of up to 5 egg chambers simultaneously. To generate a kymograph, a line was drawn along the axis of migration at the center of the anterior-posterior egg chamber axis in Fiji. In these kymographs, cell-cell interfaces are visible as lines, and their slope gives a measurement of cell migration rate. Egg chamber migration rates were calculated from the average of four-cell interface slopes from each kymograph. Egg chambers that clearly slowed down over the course of the timelapse, visible as curvature in the interface lines in the kymographs, were excluded. For an illustration of this method, see *Barlan et al., 2017*.

## Cell perimeter kymograph generation and interpretation

To visualize the distribution of Sra1-GFP along cell-cell interfaces over time, we generated kymographs of cell perimeters from timelapses of Sra1-GFP-expressing epithelia obtained using near-TIRF microscopy. Perimeters were drawn manually in Fiji in each frame with the pencil tool, and then these perimeters were used to generate kymographs in Python. Perimeters were thinned to 1 pixel and then perimeter pixels were sequenced with Python scikit-image and scipy libraries. Kymographs were generated with matplotlib. Kymograph rows were constructed by linearizing the perimeters from each frame, starting with the pixel directly above the cell centroid (the center of the trailing edge in control cells) and continuing counter-clockwise. Each row shows the fluorescence intensity of the perimeter

pixels in sequence. Cell perimeter lengths varied between frames, so kymograph row lengths varied and were aligned to their center position.

At the spatial and temporal resolution of the timelapses and corresponding kymographs, we cannot evaluate differences in the dynamics the puncta-scale WAVE complex accumulations highlighted in *Figure 8*. Instead, we focus on the 'region'-scale distribution of Sra1-GFP, and the stability of that distribution over time. The regions we refer to here are approximately the length of a cell-cell interface, with variation. Because the kymographs are generated from epithelia in which all cells express Sra1-GFP, we need additional information to identify the cell to which a region of Sra1-GFP enrichment belongs. We infer that Sra1-GFP is predominantly at leading edges in polarized, migratory epithelia based on the Sra1-GFP distribution in epithelia with mosaic Sra1-GFP expression (*Figure 5B*). Based on consistent correlation between Sra1-GFP enrichment and the presence of protrusions (*Figure 6A*, *Figure 6—figure supplement 1*), and its known role building lamellipodia as part of the WAVE complex (*Miki et al., 1998*; *Miki et al., 2000*; *Steffen et al., 2004*), we also infer that regions of Sra1-GFP enrichment belong to the cell that is protruding outward regardless of genotype. Our interpretations of Sra1-GFP enrichment patterns in movies and corresponding kymographs are made with these assumptions.

## Quantification of the stability of interface protrusion polarity

As a measurement of the stability of protrusive regions over time, we quantified the frequency with which the direction of protrusion switched across a cell-cell interface. These switching events occur when one cell and then its neighbor protrude across their shared interface, and they serve as a scoreable indicator of a change in the polarity state of the cells that bound the interface. To determine the switching frequency, we counted the number of switching events that occurred in each timelapse, and then determined an interface protrusion polarity switching rate by dividing the number of switches by the number of interfaces identified with cell and protrusion segmentation. We chose a hand-counting method because the protrusion segmentation error rate from the inclusion of retraction fibers and other sources (see Measurement of membrane protrusivity, protrusion length, and protrusion orientation) was sufficiently high that automated measurement of dynamic features of protrusions was unreliable.

## Colocalization of proteins along the leading-trailing interface

Data used for colocalization analysis were collected with 63 x/1.4 NA Plan-APOCHROMAT oil immersion objective to minimize chromatic aberration. Linescans were generated in Fiji by manually drawing a 10 pixel-wide segmented line along rows of leading-trailing interfaces at the follicle cell basal surface. At least 20 leading-trailing interfaces were included per egg chamber. For the Fat2$^{\Delta ICD}$ condition, in which the distribution of Fat2 expands beyond leading-trailing interfaces, we measured colocalization either along randomly oriented interfaces (*Figure 8E*) or leading-trailing interfaces (*Figure 8—figure supplement 1*) and obtained very similar results. Fluorescence intensities along the linescans were obtained with the PlotProfile function, which averages pixel intensities along the width of the line and reports a list of averaged values along the line's length. Spearman's correlation coefficients were calculated for each egg chamber in Python with the scipy.stats module. Failure to exactly follow leading-trailing interfaces and cusps in the segmented lines will artificially inflate the measured correlation, so we used correlation between E-cadherin-GFP (*Huang et al., 2009*) and Abi-mCherry as a negative control that is also subject to this inflation. Abi-mCherry and E-cadherin-GFP are slightly displaced from each other (anticorrelated) along the length of protrusions (the width of the linescans), but averaging across the line width collapses this displacement, resulting in measured intensity signals that are roughly uncorrelated. Spearman's correlation coefficients ± standard deviation are reported in the text. Linescans of leading-trailing interfaces were plotted using the fluorescence intensities from along the leading-trailing interfaces of two cells. Intensities from each fluorophore were rescaled between 0 and 1 and plotted with matplotlib in Python.

## Protrusion profile generation

Viewing only the F-actin channel in Fiji, we drew 1 pixel-wide lines down the length of F-actin bundles at the leading edge. Fluorescence intensities along these lines were obtained for all fluorophores with the Fiji PlotProfile function. In Python, these traces were aligned to the pixel with highest Fat2-3xGFP

or Ena-GFP intensity (*Figure 8G*, *Figure 8—figure supplement 1*). To calculate standard deviation, all traces were first rescaled individually so that their values ranged between 0 and 1. To plot 'protrusion profiles', the mean fluorescence was determined for each fluorophore at each pixel position, and then average values were rescaled between 0 and 1. Plots of protrusion profiles were generated with matplotlib.

## Movie generation

Migration motion was subtracted from several timelapse movies of migratory cells or epithelia for ease of visualization. Motion subtraction was performed using the Fiji MultiStackReg plugin 'translation' transformation [*Thévenaz et al., 1998*; control condition in *Figure 3—video 1*; *Figure 6—video 1*; *Figure 6—video 2* (part 1)] or by aligning to the centroid of a tracked cell in each frame using the scikit-image library [*Figure 6—video 2* (part 2); *Figure 7—video 1*]. Labels were added to movies in Fiji and then exported as uncompressed .avi files. These were encoded as 1080 p30 .mp4 files with H.264 (x264) video encoder using HandBrake 1.4.

## Reproducibility and statistical analysis

Visibly damaged egg chambers were excluded from all analyses. At least two biological replicates were performed for each experiment, and results confirmed to be qualitatively consistent. Each biological replicate included egg chambers pooled from multiple flies. Experiments and analysis were not randomized or performed blinded. Sample sizes were not predetermined using a statistical method. The number of biological replicates (n), statistical tests performed, and their significance can be found in figures or figure legends. Based on visual inspection, all data on which statistical tests were performed followed an approximately normal distribution, so tests assuming normalcy were used. Alpha was set to 0.05 for all statistical tests. Paired statistical tests were used for comparisons of cells of different genetic conditions within mosaic epithelia. All t-tests were two-tailed. One-sample t-tests were used when comparing a distribution of ratios to a null expectation of one. A one-way ANOVA was used when more than two conditions were compared. Welch's corrections were performed for the t-tests or ANOVAs of data plotted in *Figure 3C*, *Figure 7C*, *Figure 3—figure supplement 1A,B*, and *Figure 3—figure supplement 2*, for which the variance did not appear consistent between conditions. For post-hoc comparison tests, all pairs of conditions present in the corresponding plot were compared using post-hoc Tukey's multiple comparisons test with the following exceptions: the data plotted in *Figure 8E* and *Figure 8—figure supplement 1* were analyzed together, and all conditions were compared to Fat2-Abi and E-cadherin-Abi only, and in *Figure 5—figure supplement 2* only data from the same region (total, interface, or non-interface) was compared. For these, Šidák's multiple comparisons tests were used. For Welch's ANOVA, Dunnet's T3 multiple comparisons tests were used. p-values reported for all post-hoc tests were adjusted for multiple comparisons. All statistical tests except for the calculation of Spearman's correlation coefficients were performed in GraphPad Prism 9.

## Acknowledgements

We thank members of the Horne-Badovinac and Munro labs, Allison Zajac, Sherzod Tokamov, Ellie Heckscher, Michael Glotzer, and Carmen Williams for feedback throughout the study and comments on the manuscript. This work was supported by NIH R01 GM126047 to SHB, NIH R01 HD88831 to EM, NIH T32 HD055164 to AMW, and postdoctoral fellowships from the Chicago Fellows Program and Jane Coffin Childs Memorial Fund for Medical Research to SD.

## Additional information

### Funding

| Funder | Grant reference number | Author |
|---|---|---|
| National Institutes of Health | R01 GM126047 | Sally Horne-Badovinac |

| Funder | Grant reference number | Author |
|---|---|---|
| National Institutes of Health | R01 HD88831 | Edwin Munro |
| National Institutes of Health | T32 HD055164 | Audrey Miller Williams |
| Chicago Fellows Postdoctoral Award | | Seth Donoughe |
| Jane Coffin Childs Memorial Fund for Medical Research | | Seth Donoughe |

The funders had no role in study design, data collection and interpretation, or the decision to submit the work for publication.

### Author contributions
Audrey Miller Williams, Conceptualization, Data curation, Software, Formal analysis, Investigation, Visualization, Methodology, Writing – original draft, Writing – review and editing; Seth Donoughe, Conceptualization, Software, Formal analysis, Methodology, Writing – review and editing; Edwin Munro, Conceptualization, Writing – review and editing; Sally Horne-Badovinac, Conceptualization, Supervision, Funding acquisition, Writing – original draft, Writing – review and editing

### Author ORCIDs
Audrey Miller Williams http://orcid.org/0000-0001-6170-3365
Seth Donoughe http://orcid.org/0000-0002-4773-5739
Sally Horne-Badovinac http://orcid.org/0000-0002-0473-7451

### Decision letter and Author response
Decision letter https://doi.org/10.7554/eLife.78343.sa1
Author response https://doi.org/10.7554/eLife.78343.sa2

# Additional files

### Supplementary files
• Transparent reporting form

### Data availability
The code necessary to reproduce core aspects of data analysis, along with numerical data not included in source data files, are available at https://github.com/a9w/Fat2_polarizes_WAVE (Williams and Donoughe, 2022). Sequences of plasmids generated in this study are also available at https://github.com/a9w/Fat2_polarizes_WAVE (copy archived at swh:1:rev:0e1ee58588365bd3fba0099c-6f002993a18ec279). We will share the flies or plasmids themselves upon request to the corresponding author. Image and movie data are available from https://doi.org/10.6084/m9.figshare.20759314.v1.

The following dataset was generated:

| Author(s) | Year | Dataset title | Dataset URL | Database and Identifier |
|---|---|---|---|---|
| Williams AM, Donoughe S, Munro E, Horne-Badovinac S | 2022 | Fat2 polarizes the WAVE complex for collective cell migration | https://figshare.com/articles/journal_contribution/Fat2_polarizes_the_WAVE_complex_for_collective_cell_migration/20759314/1 | figshare, 10.6084/m9.figshare.20759314.v1 |

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
