## [Editor Report]

This paper addresses a fundamental aspect of cell migration, how the direction of cell migration is established. It links molecules involved in planar polarity to the migration machinery using quantitative imaging techniques capitalizing on the *Drosophila* genetic tool box. It adds to our growing knowledge of how collective cell migration is regulated and introduces an exciting new line of inquiry.

---

## [Decision Letter]

**Decision letter after peer review:**

Thank you for submitting your article "Fat2 polarizes the WAVE complex in trans to align cell protrusions for collective migration" for consideration by *eLife*. Your article has been reviewed by 3 peer reviewers, including Derek Applewhite as Reviewing Editor and Reviewer #1, and the evaluation has been overseen by a Reviewing Editor and Jonathan Cooper as the Senior Editor.

Essential revisions:

Broader/Mechanistic Comments:

A. How does Fat2 restrict WAVE activity in the trailing cell?

1. To test whether the proposed sequestration-based mechanism is feasible, quantitative imaging of the amount of WAVE recruitment to the membrane in wildtype cells adjacent to Fat2-perturbed vs control cells should be performed. Hypomorphs of WAVE complex with or without Fat2 perturbations would also be valuable if the authors are proposing sequestration.

2. Beyond sequestration of WAVE, another perhaps more likely scenario is that Fat2 alters the polarity of WAVE complex regulators such as PIP3 and active Rac. Examining the spatial dynamics of these molecules in addition the WAVE complex is the single addition that would most significantly strengthen the work. Probing these signaling nodes would have the additional benefit of revealing whether Fat2 has inputs to the trailing cell beyond the WAVE complex. This could be tested by analyzing these biosensors in WAVE complex null cells adjacent to control or Fat2 deficient cells.

B. What is nature of Fat2 coupling to WAVE/protrusions?

1.The colocalization of Fat2 versus WAVE is such an important point for this paper, it would be useful to firmly establish this relation. It would be helpful to show supplemental data normalizing Fat2 and WAVE to membrane or Cadherin to ensure their distributions reflect true enrichment rather than cell morphology. This seems to be the case, but it would be helpful to confirm.

Specific Comments:

(1) In general the N for some of the experiments seem low. I understand how difficult these experiments are to perform but some of the findings could be strengthened by more experiments. In Figure 2C, I'm a little concerned about the differences in sample size per condition. Sampling at least 5 egg chambers for the CK-666 treatment would make these data stronger. Additionally, Figure 2C shows the ratio of protrusive/total edges, but it would be helpful to include the total number of protrusive edges in each condition.

(2) The manuscript is short, but there was also more supplemental figures than full figures. Supplemental figures in general are not given as much weight so the impact of this story (which I think is potentially profound) is somewhat lost. Re-organization of figures would greatly strengthen the manuscript.

3) In the cases of Figure 4D and S7, which are looking at the stability of Sra1 localization, one way to quantify this could be by measuring the Pearson correlation of Sra1-3xGFP intensity between each frame of the timelapse image of the interface to the first frame. Stable Sra1-3xGFP enrichment will result in this correlation to remain relatively strong throughout time. However, unstable or transient Sra1-3xGFP will result in a dramatic decrease in correlation over time.

4) In the case of Figure 4E, cell edges could be tracked over time and stability of orientation can shown by the difference in the angle of protrusion formed at that edge between time points or some appropriate time window. Cell edges with stable protrusions would have small differences in angle over time, whereas edges with unstable protrusions will have large differences in angle.

---

## [Author Response]

Essential revisions:Broader/Mechanistic Comments:A. How does Fat2 restrict WAVE activity in the trailing cell?1. To test whether the proposed sequestration-based mechanism is feasible, quantitative imaging of the amount of WAVE recruitment to the membrane in wildtype cells adjacent to Fat2-perturbed vs control cells should be performed.

We were unsure which membrane region this request referred to—membrane at the leading edge or membrane away from the leading edge—and have addressed both possibilities below.

If this request referred to WAVE levels at the leading edge of *fat2* cells adjacent to wild-type cells, this measurement was in our original manuscript. WAVE levels are reduced at the leading edges of cells directly behind *fat2* mutant cells, regardless of the cells’ own genotype (former Figure 3E, now 5E).

We also considered that the request might refer to WAVE levels at the medial basal surface, where WAVE could redistribute if no longer recruited to the leading edge. Our original manuscript showed this analysis for *fat2* and wild-type cells, but did not include cells at clone boundaries because the small number of these cells makes it challenging to detect subtle changes in fluorescence intensity. Originally, we found a slight increase in WAVE levels at the medial basal region of *fat2* cells relative to wild-type ones, consistent with a broadening of the WAVE distribution in the absence of Fat2 (former Figure 3F, now Figure 5F). We have now measured more clones so that we can make paired comparisons of WAVE levels at the medial basal surface of cells at the clone boundaries (see diagram in Figure 5S2G), and found no significant change in WAVE levels between genotypes in these cells. With the expanded dataset, we also no longer detected significant differences in WAVE levels in wild-type or *fat2* cells away from genotype boundaries (formerly p=0.02, now p=0.08). We have updated Figure 5F and the associated results text (lines 186-196) and added Figure 5S2G to reflect these changes.

Our data make a strong case that Fat2 at the back of one cell recruits the WAVE complex to the leading edge of the cell behind to form a long-lived protrusive domain. However, whether this behavior is also sufficient to suppress protrusions from forming elsewhere along the cell perimeter is not clear. There was one line in the Discussion section of the original manuscript suggesting that Fat2 might sequester WAVE away from other sites. We have expanded this paragraph to include other ways in which protrusions could be suppressed at a distance (lines 371-375).

Hypomorphs of WAVE complex with or without Fat2 perturbations would also be valuable if the authors are proposing sequestration.

We were initially unsure of the rationale for this request and asked for clarification from Dr. Applewhite, who provided this additional explanation from an anonymous reviewer:

The authors are proposing that Fat2 limits protrusion formation in adjacent cells through WAVE complex sequestration. Under this model, the increased protrusion formation that accompanies loss of Fat2 is a consequence of the increased cytosolic pool of WAVE complex. So if Fat2 normally sequesters say 50% of the WAVE complex, then a WAVE complex heterozygote (loss of 50% of protein) should rescue the enhanced protrusion elicited by loss of Fat2. In contrast, if Fat2 is playing a role in regulating cell polarity beyond WAVE sequestration, then adjusting the WAVE dose to compensate for the amount normally recruited to Fat2 should have no effect.

As noted in point 1, the manuscript has been modified to no longer propose a sequestration based mechanism. It should also be noted that, although the WAVE complex and protrusions are more widely distributed around the basal cell perimeter in the absence of Fat2, their overall levels are reduced when compared to the wild-type condition, not increased (Figures 3A-C, 5S2C). Given this relationship, further reducing WAVE levels in *fat2* mutant epithelia seems unlikely to shed light on Fat2’s mechanism of action.

2. Beyond sequestration of WAVE, another perhaps more likely scenario is that Fat2 alters the polarity of WAVE complex regulators such as PIP3 and active Rac. Examining the spatial dynamics of these molecules in addition the WAVE complex is the single addition that would most significantly strengthen the work. Probing these signaling nodes would have the additional benefit of revealing whether Fat2 has inputs to the trailing cell beyond the WAVE complex. This could be tested by analyzing these biosensors in WAVE complex null cells adjacent to control or Fat2 deficient cells.

We agree that Fat2 could act more directly on PIP3 or Rac nodes of the lamellipodia circuit, and thereby polarize WAVE, and we appreciate the reviewer’s suggestion to delve into this. To evaluate Rac polarization, we imaged the FRET sensor reported by the Montell lab (doi:10.1038/ncb2061, BDSC:32050), expressed either in all follicle cells or mosaically, allowing us to distinguish leading edge signal from that of the adjacent trailing edge. We measured the FRET ratio using the approach described here: (doi:10.1038/nprot.2011.395) after seeking advice from a senior postdoc in the Montell lab. However, we were unable to detect any FRET signal polarization. We likewise looked for polarized PI(3,4,5)P3 using a Grp1-PH-based probe (tubP>GFP-Grp1-PH: BDSC8163), but were again unable to detect polarized signal—in this case the probe had the same distribution as a plasma membrane label. Unfortunately, these are the only two biosensors that are currently readily available for *Drosophila*. We think it is highly likely that active Rac and PIP3 are polarized in this system, but we are unable to confirm this or test the role of Fat2 in that polarization with the accessible tools.

Our initial submission included a paragraph in the Discussion noting the possibility that Fat2 could act through Rac or PIP3, but this came late in the manuscript. We have now modified our Results section to emphasize that we are using WAVE complex localization as a readout for the entire circuit (lines 156-161).

B. What is nature of Fat2 coupling to WAVE/protrusions?1.The colocalization of Fat2 versus WAVE is such an important point for this paper, it would be useful to firmly establish this relation. It would be helpful to show supplemental data normalizing Fat2 and WAVE to membrane or Cadherin to ensure their distributions reflect true enrichment rather than cell morphology. This seems to be the case, but it would be helpful to confirm.

To control for effects of morphology on the apparent colocalization between Fat2 and WAVE, we measured the colocalization between Abi-mCherry and E-cadherin-GFP, which is near-uniformly distributed along interfaces. Abi and E-cadherin show a positive Spearman’s correlation, indicating that interface morphology may contribute to positive Fat2-WAVE correlation (Figure 8S1B). But the correlation between Fat2 and the WAVE complex is significantly higher than this negative control in all backgrounds we measured, showing that Fat2 and WAVE do colocalize (Figure 8S1B). These data were present in our original submission, but we only discussed them as a control for our region selection method. We have now added text in the Results section stating that these data also serve as a control for cell morphology (lines 261-262).

Specific Comments:1) In general the N for some of the experiments seem low. I understand how difficult these experiments are to perform but some of the findings could be strengthened by more experiments. In Figure 2C, I'm a little concerned about the differences in sample size per condition. Sampling at least 5 egg chambers for the CK-666 treatment would make these data stronger.

We have now collected a total of 5 CK-666 timelapse videos and reanalyzed membrane protrusion data with this expanded dataset (see Figures 3, 3S1).

Additionally, Figure 2C shows the ratio of protrusive/total edges, but it would be helpful to include the total number of protrusive edges in each condition.

We have added a supplemental table (Figure 3-source data 1) that includes these values. The table is referenced in the legend of Figure 2C (now Figure 3C).

2) The manuscript is short, but there was also more supplemental figures than full figures. Supplemental figures in general are not given as much weight so the impact of this story (which I think is potentially profound) is somewhat lost. Re-organization of figures would greatly strengthen the manuscript.

We have reorganized the manuscript so that data from 3 formerly supplemental figures are now included in the main figures (current Figures 2,4,6EF). There are now a total of 8 main figures and 6 supplemental figures (formerly 5 main, 9 supplemental).

3) In the cases of Figure 4D and S7, which are looking at the stability of Sra1 localization, one way to quantify this could be by measuring the Pearson correlation of Sra1-3xGFP intensity between each frame of the timelapse image of the interface to the first frame. Stable Sra1-3xGFP enrichment will result in this correlation to remain relatively strong throughout time. However, unstable or transient Sra1-3xGFP will result in a dramatic decrease in correlation over time.

We agree that quantification of Sra1 dynamics would strengthen the manuscript. However, the migratory state of the tissue confounds many measurements of these types of subcellular dynamics. Because the entire tissue is migrating in control, but not *fat2*, epithelia, we cannot use simple intensity correlation over time to compare Sra1 subcellular dynamics. We attempted to register the migrating cells to isolate the contribution of within-cell Sra1 distribution changes, but control cells still tend to deform more over time, so this didn’t fully remove the confound. As an imperfect stand-in for quantification, we have included a video showing Sra1 dynamics in a field of control and *fat2* cells, allowing the reader to assess the generality of *fat2* domain fluctuations among tens of cells in a shared epithelium. We have also added a quantification of membrane protrusion stability (see point 9). Sites of Sra1 enrichment correspond closely with sites of membrane protrusion (see Figures 6A, 6S1), so we can strongly infer that the Sra1 distribution is reflected in the membrane protrusion distribution.

4) In the case of Figure 4E, cell edges could be tracked over time and stability of orientation can shown by the difference in the angle of protrusion formed at that edge between time points or some appropriate time window. Cell edges with stable protrusions would have small differences in angle over time, whereas edges with unstable protrusions will have large differences in angle.

We have now added a measurement of the stability of membrane protrusion distribution, in which we quantify the frequency of events in which the direction of protrusion from an individual interface switches. We chose this measurement because, as a binary event, it was more readily score-able than the related quantity of changes in net cell polarity exemplified in Figure 7A. The frequency of polarity switches per interface per hour was significantly higher in *fat2* epithelia than controls (Welch’s t-test, p=0.0002). These data are now plotted in Figure 7C with an accompanying example of a switching event in Figure 7B.

Ideally, we would have used our membrane protrusion segmentation and polarity data as a starting point for measuring protrusion stability. However, although the segmentation was reliable enough to detect a bias in protrusion polarity shown in the rose plots, there are errors. These errors have an outsized impact on measurements of traits with a time dimension like stability because they depend on successful segmentation of a protrusion for many consecutive frames. Measurements of protrusion orientation stability over time were strongly influenced by these errors. For this reason, we manually counted all polarity switches in the new analysis.